# Biophysical K$_v$3 channel alterations dampen excitability of cortical PV interneurons and contribute to network hyperexcitability in early Alzheimer's

Viktor J Olah[1†], Annie M Goettemoeller[1,2†], Sruti Rayaprolu[3], Eric B Dammer[4], Nicholas T Seyfried[4], Srikant Rangaraju[3], Jordane Dimidschstein[5], Matthew JM Rowan[1]*

[1]Department of Cell Biology, Emory University, Atlanta, United States; [2]GDBBS Graduate Program, Laney Graduate School, Emory University, Atlanta, United States; [3]Department of Neurology, Emory University, Atlanta, United States; [4]Department of Biochemistry, Emory University, Atlanta, United States; [5]Stanley Center for Psychiatric Research, Broad Institute, Cambridge, United States

**\*For correspondence:**
mjrowan@emory.edu

[†]These authors contributed equally to this work

**Competing interest:** The authors declare that no competing interests exist.

## Abstract

In Alzheimer's disease (AD), a multitude of genetic risk factors and early biomarkers are known. Nevertheless, the causal factors responsible for initiating cognitive decline in AD remain controversial. Toxic plaques and tangles correlate with progressive neuropathology, yet disruptions in circuit activity emerge before their deposition in AD models and patients. Parvalbumin (PV) interneurons are potential candidates for dysregulating cortical excitability as they display altered action potential (AP) firing before neighboring excitatory neurons in prodromal AD. Here, we report a novel mechanism responsible for PV hypoexcitability in young adult familial AD mice. We found that biophysical modulation of K$_v$3 channels, but not changes in their mRNA or protein expression, were responsible for dampened excitability in young 5xFAD mice. These K$^+$ conductances could efficiently regulate near-threshold AP firing, resulting in gamma-frequency-specific network hyperexcitability. Thus, biophysical ion channel alterations alone may reshape cortical network activity prior to changes in their expression levels. Our findings demonstrate an opportunity to design a novel class of targeted therapies to ameliorate cortical circuit hyperexcitability in early AD.

## Editor's evaluation

Using computational modeling and dynamic clamp recordings, this work supports the concept that hyperexcitability of cortical circuits in a familial mouse model of Alzheimer's disease is caused by impairments of biophysical properties of K$_v$3 channels in parvalbumin-positive cortical interneurons.

## Introduction

Unraveling mechanisms that initiate cognitive decline in Alzheimer's disease (AD) is a central aim in neuroscience. A prevailing model of AD posits that progressive deposition of toxic protein aggregates sparks a neuropathological cascade. However, recent work suggests that early cognitive dysfunction is uncoupled from these aggregates (*Arroyo-García et al., 2021*; *Nuriel et al., 2017*; *Shimojo et al., 2020*). Several alternative models for early cognitive decline are under consideration (*De Strooper and Karran, 2016*; *Frere and Slutsky, 2018*), including abnormal circuit activity (*Busche and Konnerth, 2015*; *Busche et al., 2008*; *Cirrito et al., 2005*; *Davis et al., 2014*; *Pooler et al.,*

*2013*; *Wu et al., 2016*). Circuit hyperexcitability is evident in several mouse models of familial (FAD) and sporadic AD (*Lamoureux et al., 2021*; *Minkeviciene et al., 2009*; *Nuriel et al., 2017*), including at prodromal stages (*Bai et al., 2017*; *Busche and Konnerth, 2015*). Furthermore, abnormal brain activity is apparent in humans with mild cognitive impairment (*Dickerson et al., 2005*; *Hämäläinen et al., 2007*; *Miller et al., 2008*; *Sperling et al., 2010*) and in early FAD (*Quiroz et al., 2010*; *Sepulveda-Falla et al., 2012*). These shifts in circuit activity may result from dysfunctional neuronal firing and neurotransmission (*Chen et al., 2018*; *Palop and Mucke, 2016*). However, the cellular and molecular mechanisms underlying these neuronal deficits are not yet fully understood.

Cognition and memory require carefully balanced excitatory and inhibitory activity (*Zhou and Yu, 2018*). In different AD mouse models, impairments in inhibition precede plaque formation, disrupting brain rhythms associated with memory formation (*Arroyo-García et al., 2021*; *Li et al., 2021*; *Nuriel et al., 2017*; *Sederberg et al., 2006*). Modified inhibitory tone in early AD is likely related to changes in the intrinsic excitability of local circuit inhibitory interneurons. For example, AP firing is altered in 'fast spiking' PV interneurons in different human APP (hAPP)-expressing mice (*Arroyo-García et al., 2021*; *Caccavano et al., 2020*; *Chen et al., 2018*; *Martinez-Losa et al., 2018*; *Petrache et al., 2019*; *Verret et al., 2012*). Interestingly, altered PV physiology may occur before changes to other neighboring neuron subtypes (*Hijazi et al., 2020*; *Park et al., 2020*). Altered AP firing in PV cells could result from changes in the expression of genes that regulate excitability (*Martinez-Losa et al., 2018*). However, major shifts in gene and protein expression may only materialize after substantial plaque formation (*Bundy et al., 2019*) in AD. Thus, a systematic evaluation of molecular mechanisms contributing to altered firing in PV cells is required.

In this study, we used a viral-tagging method to examine PV interneuron excitability in the somatosensory cortex of young adult 5xFAD mice. PV interneurons from 5xFAD mice displayed strongly dampened firing near-threshold and modified action potential (AP) waveforms, indicating dysregulation of either $Na^+$ or $K^+$ channels. Combined examination of several AP firing parameters, computational modeling, and PV-specific qPCR indicated that changing $Na^+$ channel availability was not responsible for changes in AP firing. However, we observed alterations in $K^+$ channel activation and kinetics in AD mice, independent of changes in $K^+$ gene expression. Using dynamic clamp and additional PV modeling, we found that these shifts in $K^+$ channel activation could recapitulate the observed phenotypes in 5xFAD mice. Furthermore, $K^+$ channel-induced changes in PV firing were sufficient to induce circuit hyperexcitability and modified gamma output in a reduced cortical model. Together, these results establish a causal relationship between ion channel regulation in PV interneurons and cortical circuit hyperexcitability in early AD, independent of changes in gene expression.

## Results

### Near-threshold suppression of AP firing in PV interneurons of young 5xFAD mice

To evaluate physiological phenotypes of PV interneurons in 5xFAD and wild-type (WT) control mice, we implemented an AAV viral-enhancer strategy (*Vormstein-Schneider et al., 2020*) to specifically label PV interneurons. Mature animals were injected with this PV-specific vector (referred throughout as 'AAV.E2.GFP') in layer 5 somatosensory cortex before plaque formation (postnatal days 42–49) (*Bundy et al., 2019*; *Jawhar et al., 2012*; *Li et al., 2021*; *Oakley et al., 2006*). Acute slices were obtained ~7 days later, and GFP-expressing (GFP$^+$) cells were targeted for patch clamp using combined differential contrast and epifluorescent imaging (*Figure 1A*). Current-clamp recordings from WT mice displayed high-frequency, nonadaptive repetitive spiking characteristics of PV cells (*Figure 1B*). In addition, the expression of several known PV interneuron genes was confirmed in AAV. E2.GFP$^+$ neurons (*Chow et al., 1999*; *Ogiwara et al., 2007*; *Rudy et al., 2011*) using qPCR, the levels of which were indistinguishable from PV interneurons isolated in an identical fashion from PV-Cre mice (*Figure 1—figure supplement 1*).

Recent studies of several different hAPP-expressing mouse models have demonstrated abnormal AP firing in GABAergic interneurons at different stages of plaque deposition (*Hijazi et al., 2020*; *Mondragón-Rodríguez et al., 2018*; *Park et al., 2020*; *Petrache et al., 2019*; *Verret et al., 2012*). In prodromal 5xFAD mice, we found that continuous spiking was severely dampened in layer 5 PV neurons in the near-threshold range; however, spike frequency was unaltered near their maximal firing

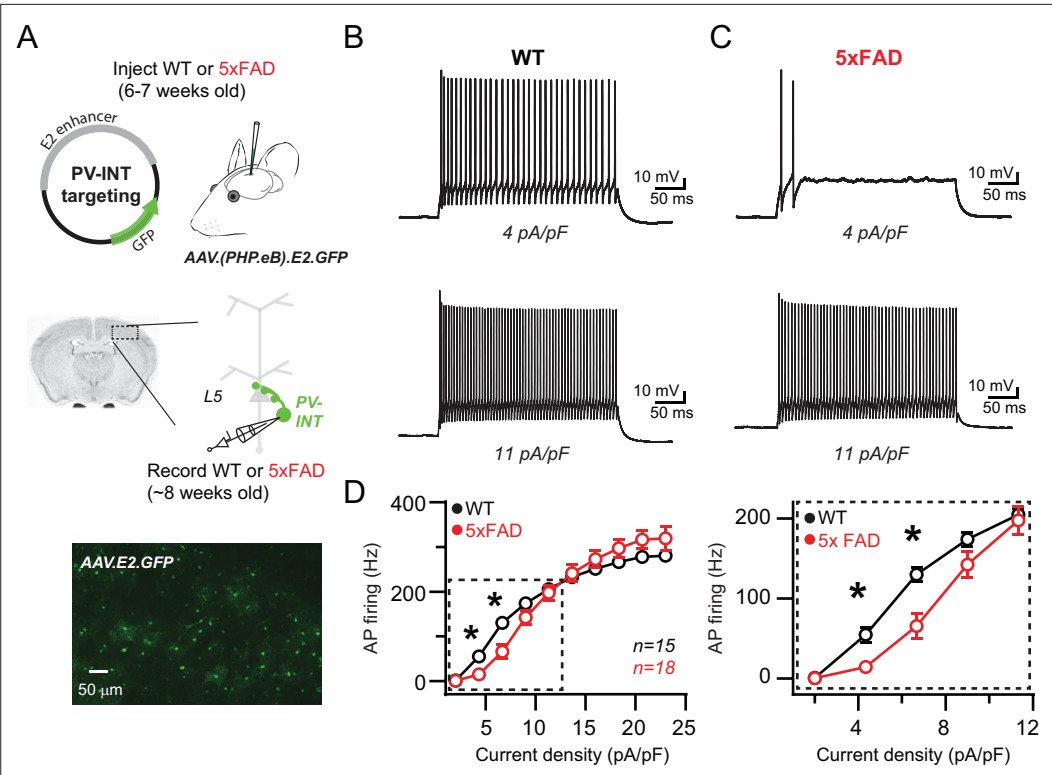

**Figure 1.** Reduced action potential (AP) firing frequency in parvalbumin (PV) interneurons of young 5xFAD mice. (**A**) Graphical summary of AAV.E2.GFP stereotactic injection in somatosensory cortex and subsequent whole-cell current-clamp recordings from GFP[+] PV interneurons (PV-INT). (**B**) AP firing elicited in wild-type (WT) mice by square pulse current injections of varying magnitude normalized to cellular capacitance during recordings. (**C**) AP firing elicited in 5xFAD mice at current density levels matched to WT mice for comparison. (**D**) Group data summary of AP firing frequency in WT and 5xFAD mice. Significance was defined by repeated-measures two-way ANOVA (p<0.05) with Sidak's multiple-comparison test. For all summary graphs, data are expressed as mean (± SEM).

The online version of this article includes the following figure supplement(s) for figure 1:

**Figure supplement 1.** Confirmation of parvalbumin (PV) interneuron gene expression in AAV.E2.GFP[+] neurons.

rate (*Figure 1C and D*). Passive parameters were unaltered when comparing WT and 5xFAD, including input resistance (94.9 ± 5.9 and 103.5 ± 8.4 MΩ; p=0.83; unpaired *t*-test) and holding current immediately after break-in (17.5 ± 7.8 and 19.1 ± 10.5 pA; measured at –60 mV; p=0.41; unpaired *t*-test), suggesting that an active mechanism was responsible for the observed differences in spike frequency.

## Altered AP waveform and excitability are uncoupled from changes in Na_v channels properties and mRNA expression

The extraordinarily rapid onset and repolarization of PV-APs depends on the combined expression of fast voltage-gated sodium (Na_v) and potassium (K_v) channel families (*Baranauskas et al., 2003*; *Catterall et al., 2010*; *Cheah et al., 2012*; *Erisir et al., 1999*; *Goldberg et al., 2008*; *Gu et al., 2018*; *Rudy and McBain, 2001*; *Wang et al., 1998*). Whether altered expression of voltage-gated channels emerges before plaque deposition is unclear. Changes in the expression of channels from the Na_v1 family may contribute to altered spiking in cortical PV interneurons from hAPP-expressing FAD mice (*Martinez-Losa et al., 2018*; *Verret et al., 2012*; but see *Saito et al., 2016*). Therefore, we examined parameters associated with fast-activating Na_v channels (*Kole et al., 2008*; *Li et al., 2014*; *Platkiewicz and Brette, 2010*); however, we found no significant differences between 5xFAD and control mice (*Figure 2A and B*). AP afterhyperpolarization (AHP) amplitude was also unaltered (*Figure 2B*).

Na_v channel deficits result in reduced AP amplitude and contribute to AP failure during repetitive firing (*Catterall et al., 2010*; *Escayg and Goldin, 2010*; *Gu et al., 2018*; *Van Wart and Matthews,*

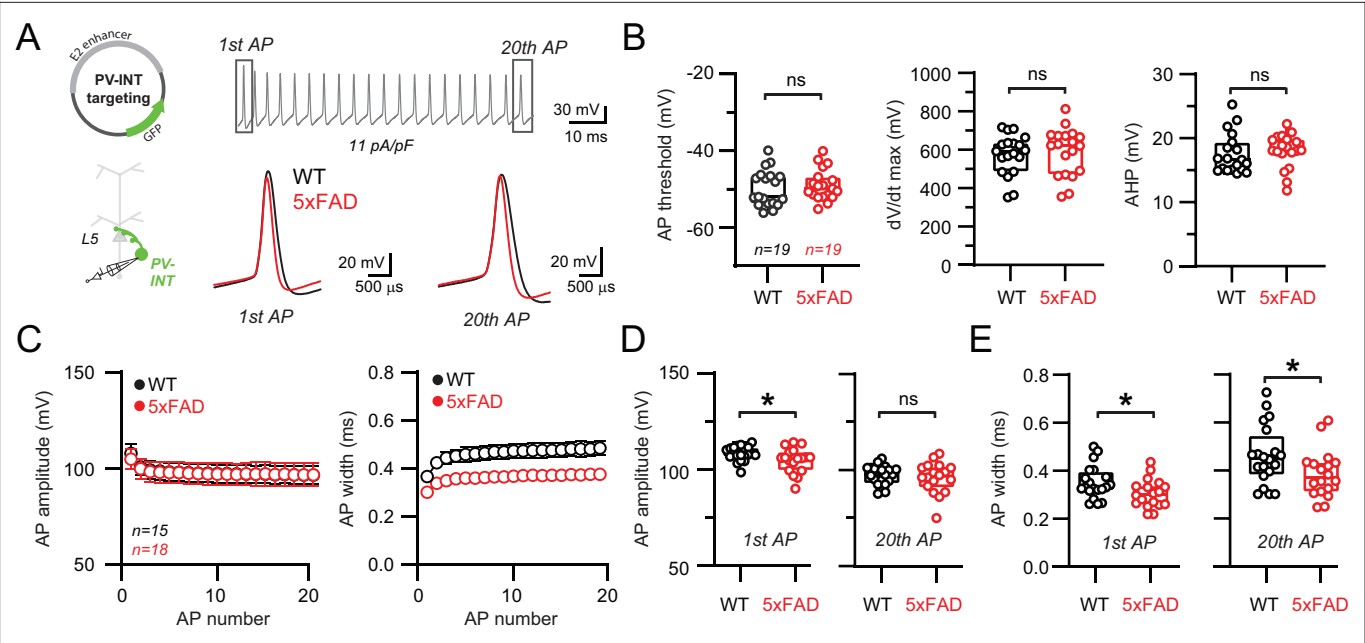

**Figure 2.** Altered action potential (AP) waveforms in parvalbumin (PV) interneurons of 5xFAD mice. (**A**) AP waveforms and properties of GFP+ interneurons were compared at 11 pA/pF square pulse injections in wild-type (WT) and 5xFAD mice. In the enlarged view, APs from the 1st and 20th spike in the train of WT and 5xFAD mice are superimposed for comparison. (**B**) Summary data of AP properties. No differences in AP threshold, dV/dt maximum, or afterhyperpolarization (AHP) were observed (p>0.05; unpaired *t*-test). (**C**) Relationship between AP amplitude or width in WT and 5xFAD mice and AP # during spike trains elicited with 11 pA/pF current injection. Data are expressed as mean (± SEM). (**D**) Summary data of AP amplitude for the 1st and 20th APs in WT and 5xFAD mice. (**E**) Summary data of AP width for the 1st and 20th APs in WT and 5xFAD mice. For (**B, D, E**), individual data points and box plots are displayed. Significance is defined as p<0.05; unpaired *t*-tests.

*2006*). Using a serendipitous current injection step where spike frequency was indistinguishable between 5xFAD and control mice (11 pA/pF; *Figure 2A*), a subtle reduction in the amplitude of the initial AP was observed (*Figure 2D*). However, this reduction did not progressively worsen during continued firing (*Figure 2C and D*) as seen in mouse models where $Na_v1$ channels were altered (*Ogiwara et al., 2007*; *Yu et al., 2006*). Interestingly, AP repolarization was more rapid across the entire spike train (quantified as a reduction in full AP width at half-maximal amplitude [half-width]; *Figure 2C and D* ) in 5xFAD mice.

To test whether an $Na_v$ channel mechanism could describe the AP firing phenotypes observed in 5xFAD mice, we built a simplified PV NEURON model constrained by our measurement AP parameters. Using the model, we independently simulated how changes in overall $Na_v$ conductance, activation voltage, and kinetic properties affected relevant AP firing properties (*Figure 3A*). Significant reduction of $Na_v$ conductance density (up to 50% of control) could lessen AP firing at near-threshold current steps (*Figure 3B*). However, this reduction was accompanied by complete firing failures at high frequencies (*Verret et al., 2012*; *Figure 3B*), which was not observed in 5xFAD mice. Furthermore, AP width was unaltered over a broad range of $Na_v$ conductance densities (*Figure 3C*), suggesting that AP width narrowing observed in 5xFAD mice was also due to an $Na_v$-independent mechanism. In contrast, changing $Na_v$ conductance density was associated with changes in AP threshold and maximal dV/dt (*Figure 3D*), which were unaltered in our recordings (*Figure 2*). Shifting $Na_v$ kinetics or activation voltage also could not explain the observed 5xFAD phenotypes (*Figure 3—figure supplement 1*).

To complement our $Na_v$ modeling, we performed PV interneuron-specific qPCR (*Tasic et al., 2018*) by isolating and pooling AAV.E2.GFP+ neurons from dissected somatosensory cortex following AAV retro-orbital injection (*Chan et al., 2017*) in 5xFAD and control mice (*Figure 3E*). Expression of $Na_v1.1$ (*Scn1a*) and $Na_v1.6$ (*Scn8a*) was detected in WT PV interneurons (*Figure 3F*). Relative to control, no changes in mRNA expression of either subunit in 5xFAD mice were found (*Figure 3G*). Together, our patch-clamp recordings, simulations, and gene expression data indicate that modifications in $Na_v$ channel expression cannot account for the observed changes in PV firing in our pre-plaque hAPP model.

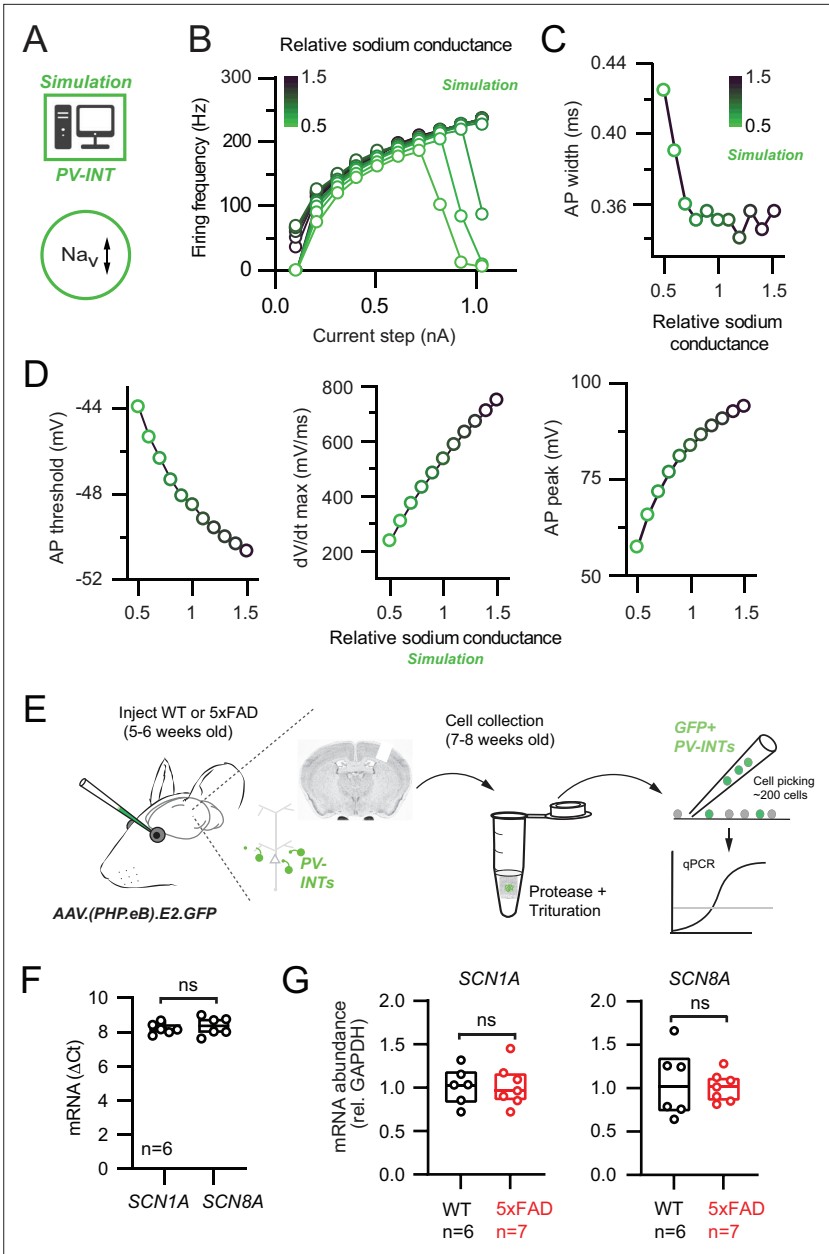

**Figure 3.** Na$_v$ channel changes do not explain changes in parvalbumin (PV) interneuron excitability in 5xFAD mice. (**A**) Depiction of PV cell single-compartmental model with modified Na$_v$ channel properties. (**B**) Simulated relationship (S/cm$^2$) between the magnitude of injected current and action potential (AP) firing frequency at variable Na$_v$ conductance densities. (**C**) Summary relationship of AP width and relative Na$_v$ conductance density (±50% from control Na$_v$ conductance). (**D**) Summary graphs depicting the effect of changing Na$_v$ conductance density on AP threshold, dV/dt maximum, and AP peak (±50% from control Na$_v$ conductance). (**E**) Depiction of cell-type-specific qPCR of *Scn1* genes following retro-orbital AAV injection in 4–6-week-old mice. Individual neurons were physically isolated, hand-picked, and pooled after allowing 2–3 weeks for cortical expression. (**F**) Comparative qPCR expression of *Scn1a* and *ScnN8a* in wild-type (WT) mice. (**G**) Quantification of *Scn1a* and *Scn8a* mRNA expression between WT and 5xFAD mice. For (**F**) and (**G**), data are expressed as individual data points from each individual mouse with box plots superimposed.

The online version of this article includes the following figure supplement(s) for figure 3:

**Figure supplement 1.** Na$_v$ channel changes do not explain changes in parvalbumin (PV) interneuron excitability in 5xFAD mice.

## Biophysical but not gene expression changes of K$_v$3 channels in PV interneurons

The distinct firing phenotype and rapid AP repolarization of fast-spiking PV cells require expression of fast-activating K$_v$ channels, which complement Na$_v$1 (**Gu et al., 2018**). Thus, by ruling out Na$_v$ channels as viable candidates for explaining the above differences, we postulated that altered K$_v$ channel availability could contribute to AP firing differences observed in 5xFAD mice. Tetraethylammonium (TEA)-sensitive K$_v$3 channels are highly expressed in PV cells and possess extremely fast kinetics that set AP width and firing rate in different neuron types (**Barry et al., 2013**; **Erisir et al., 1999**; **Rowan et al., 2014**; **Song et al., 2005**). To record K$_v$ conductances from PV interneurons, we obtained outside-out patches from AAV.E2.GFP$^+$ neurons in both 5xFAD and control mice. TEA (1 mM) was puffed onto isolated patches to block and *post-hoc* evaluation (**Figure 4A**).

Large TEA-sensitive currents were isolated in patches from PV cells (**Figure 4B**) displaying characteristic K$_v$3-like properties, including a relatively depolarized steady-state half-activation voltage (**Figure 4D**) and submillisecond activation kinetics (**Figure 4E**; **Baranauskas et al., 2003**; **Lien et al., 2002**; **Rudy and McBain, 2001**). Substantial changes in K$_v$ channel availability could account for the observed differences in AP firing in 5xFAD mice (**Figure 1**). However, the overall TEA-subtracted conductance was unchanged in 5xFAD (**Figure 4C**), suggesting that overall K$_v$ channel surface expression was unaltered. The proportion of TEA-insensitive conductance was also unchanged (WT, 33.1% ± 2.9%; 5xFAD, 33.0% ± 2.3%; p=0.98; unpaired *t*-test; n = 9 and 12; respectively). Interestingly, we observed differences in the biophysical properties of TEA-sensitive channels in 5xFAD. Channels activated at more hyperpolarized (left-shifted) voltages (**Figure 4D**; half-activation voltage –6.6 mV WT vs. –15.5 mV in 5xFAD). Furthermore, activation kinetics decreased across the observable range in 5xFAD mice (**Figure 4E**). We also performed recordings to evaluate steady-state inactivation parameters and kinetics (**Figure 4—figure supplement 1A–C**). On average, voltage dependence of activation and inactivation from WT recordings was in agreement with the biophysical characteristics of K$_v$3.3 channels (**Fernandez et al., 2003**). Inactivation kinetics were highly variable, but on average resembled K$_v$3.3 homomers (**Weiser et al., 1994**) or K$_v$3.1/K$_v$3.4 heteromers (**Baranauskas et al., 2003**), but other possible compositions cannot be excluded. In contrast to changes in K$_v$ activation voltage in 5xFAD, half inactivation voltage was slightly right-shifted (half inactivation voltage –19.9 mV in WT vs. –13.9 mV in 5xFAD). Inactivation kinetics were indistinguishable in WT and 5xFAD (**Figure 4—figure supplement 1B and C**).

Differential mRNA expression of the four known K$_v$3 channel *Kcnc* subunits in 5xFAD mice could account for the observed shifts in K$_v$3 biophysics (**Figure 4D and E**). To evaluate this possibility, we again performed PV interneuron-specific qPCR by isolating AAV.E2.GFP$^+$ cells (**Figure 4F**), as described earlier. Expression of all four subunits was confirmed in PV cells from somatosensory cortex; however, no differences in mRNA expression were found between 5xFAD and control mice for any of the four *Kcnc* subunits (**Figure 4F**). Several studies have demonstrated a discordance between steady-state mRNA and protein levels (**de Sousa Abreu et al., 2009**; **Vogel and Marcotte, 2012**). To evaluate whether altered protein levels of ion channels could account for AP firing differences in young 5xFAD mice, we examined quantitative mass spectrometry data related to K$^+$ and Na$^+$ channel proteins obtained from cortical homogenates from WT and 5xFAD mice (1.8, 3.1, 6.0, 10.2, and 14.4 months old). Protein levels from nearly all *Kcnc* (K$_v$3), *Kcna* (K$_v$1), *Kcnq* (K$_v$7), *Kcnd* (K$_v$4), *Kcnma1* (BK$_{Ca2+}$), and *Scn1* (Na$_v$1) subunits, as well as other K$^+$ and Na$^+$ channel families and regulatory subunits, were quantified (**Figure 4—figure supplement 2**).

K$_v$3 protein (K$_v$3.1, 3.2, and 3.3) levels at the youngest timepoint (7.2 weeks old), which matched our earlier physiological and mRNA evaluations, were again unchanged, while Na$_v$1.1 was slightly increased (**Figure 4—figure supplement 2B**; 5.9%; unadjusted p<0.05) in 5xFAD. Protein levels for most other examined channel types and regulatory subunits were unaltered in young 5xFAD mice (**Figure 4—figure supplement 2B and C**). However, several age-related trends were noted. After showing a slight increase in young 5xFAD mice, Na$_v$1.1 levels were reduced at 10.2 months old (**Figure 4—figure supplement 2E**). Additionally, K$_v$3.3 levels progressively reduced with age (**Figure 4—figure supplement 2B–F**). In general, proteomic alterations expanded with increasing age in 5xFAD mice (see **Figure 4—source data 1**). Protein levels of PV and CaMKII were unchanged at 1.8- and 3-month timepoints (p>0.05; one-way ANOVA). Together, our combined mRNA and protein-level evaluations indicate that the modifications responsible for divergent K$_v$ biophysical properties occur without changes in mRNA or protein levels at this pre-plaque disease stage.

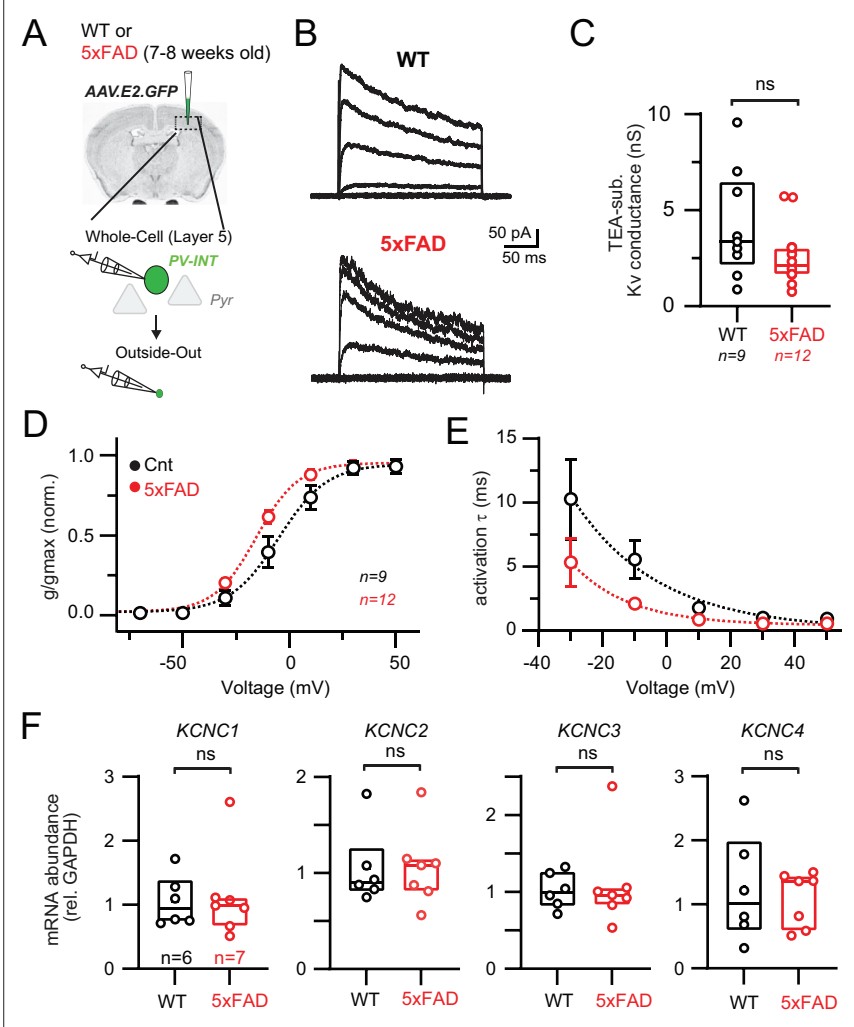

**Figure 4.** Modified K$_v$3 channel biophysics in 5xFAD mice. (**A**) Experimental workflow for obtaining outside-out patches from parvalbumin (PV) interneurons in wild-type (WT) and 5xFAD mice. (**B**) Representative K$_v$3 currents isolated from outside-out patches in WT and 5xFAD mice. Patches were held at –110 mV and then stepped from –90 to +50 mV using 300 ms, 20 mV steps. (**C**) Data summary of maximal K$_v$3 conductance in WT and 5xFAD mice ($p > 0.05$; unpaired $t$-test). Individual data points from each patch and box and whisker plot summaries are displayed. (**D**) Summary of activation voltage of K$_v$3 conductance isolated from patches in WT and 5xFAD mice. Conductance was normalized to the maximal overall conductance (gmax) for each cell. The average dataset was fit with a Boltzmann function with individual values expressed as mean (± SEM). (**E**) Summary of activation time constant ($\tau$) of K$_v$3 currents in isolated from patches in WT and 5xFAD mice. Datasets were fit with single monoexponential decay functions and are expressed as mean (± SEM). (**F**) Comparison summary of *Kcnc*1-4 mRNA expression between WT and 5xFAD mice from isolated and pooled PV interneurons. Individual data points from each mouse and box plot summaries are displayed. No differences were found between WT and 5xFAD cohorts for any of the four subunits ($p > 0.05$; unpaired $t$-tests).

The online version of this article includes the following source data and figure supplement(s) for figure 4:

**Source data 1.** Mass spectrometry of protein levels in 5xFAD mice.

**Figure supplement 1.** Observed K$_v$3 inactivation properties and relationship to action potential (AP) firing in parvalbumin (PV) interneurons.

**Figure supplement 2.** Mass spectrometry (Mass Spec) of protein levels at varying ages in 5xFAD mice.

## Modified K$_v$3 channel biophysics recapitulate the 5xFAD phenotypes in a PV model

To test whether modifying K$_v$3 channel biophysics alone could adequately explain the AP firing phenotypes in 5xFAD mice, we returned to our reduced PV cell simulation (*Figure 5A*). In control conditions, our model PV neuron increased firing in relation to the magnitude of current injection (*Figure 5B and C*). Notably, when the K$_v$3 activation potential dependence was hyperpolarized as observed in 5xFAD PV neurons (*Figure 5A and B*; control absolute half-activation voltage = –5.0 mV; absolute test Vshift (–10 mV)=–15.0 mV), we found that AP firing was strongly dampened in the near-threshold range (*Figure 5B and C*; see also *Lien and Jonas, 2003*), mirroring changes in 5xFAD mice. This near-threshold reduction in firing remained stable at differing inactivation voltage dependences (*Figure 4—figure supplement 1E*). Shifting the K$_v$3 activation voltage leftward also led to a slight reduction in firing frequency at higher current injection levels, which could be normalized with a concurrent increase in K$_v$3 activation kinetics (tau) (*Figure 5C*).

Modulation of K$_v$3 activation kinetics alone could modify AP firing frequency in either direction (*Figure 5D*), likely owing to changing Nav channel use dependence. In contrast, broadly shifting K$_v$3 inactivation kinetics had no effect on either near-threshold or saturating firing frequencies (*Figure 4—figure supplement 1F and G*). This is likely because extremely rapid PV-APs (half-width ~350 µs) are too brief for K$_v$3 inactivation to accumulate, even with very rapid (tau = 50 ms) inactivation kinetics (*Figure 4—figure supplement 1G*). AP repolarization is differentially shaped by distinct kinetic properties of different K$_v$ subtypes (*Bean, 2007*; *Dodson et al., 2002*; *Pathak et al., 2016*; *Rowan et al., 2014*; *Wang et al., 1998*). As AP width in our PV cell model was uncoupled from changes in Na$_v$ conductance, we hypothesized that AP width was influenced by changes in K$_v$3 channel kinetics (*Baranauskas et al., 2003*). Indeed, increased activation kinetics were correlated with a reduction in AP width, which could also influence AP amplitude (*Figure 5E*). In contrast, changes in K$_v$3 inactivation kinetics had no effect on AP width or amplitude (*Figure 4—figure supplement 1H*).

Other potassium channel types may also be sensitive to 1 mM TEA and thus contribute to biophysical alterations in patches from 5xFAD mice, in particular, B$_K$ and K$_v$7.2 (*Coetzee et al., 1999*). When expressed locally, B$_K$ channels can influence AP repolarization (*Alle et al., 2011*; *Casale et al., 2015*). However, B$_K$ blockade in PV-expressing interneurons in cortex or cerebellum did not affect AP width (*Casale et al., 2015*; *Rowan et al., 2014*) or spike frequency, likely due to functional confinement of B$_K$ to axonal synapses in PV cells (*Erisir et al., 1999*; *Goldberg et al., 2005*). These factors suggest that TEA-sensitive currents isolated in outside-out patches in this study unlikely to include B$_K$. To confirm this, we puffed iberiotoxin (IBTX) (*Casale et al., 2015*; *Goldberg et al., 2005*) onto outside-out patches from layer 5 PV interneurons. No changes in outward conductance were identified following IBTX (control, 5.2 ± 1.6 nS; IBTX, 5.1 ± 1.5 nS; p>0.05, paired *t*-test; n = 5), indicating the absence of active B$_K$ conductance in our patch recordings.

Although Kv7 kinetics are likely not rapid enough to regulate AP width, if present, subthreshold activation of K$_v$7 could contribute to changes in AP firing (*Guan et al., 2011*) in 5xFAD mice. Therefore, we supplemented our original K$_v$3 model with an additional K$_v$7 conductance (*Sekulić et al., 2015*; *Figure 5—figure supplement 1*). Addition of K$_v$7 (2 mS/cm$^2$) could reduce firing across a range of current injections (*Figure 5—figure supplement 1B and C*). However, in contrast to K$_v$3 (*Figure 5B*), hyperpolarizing the K$_v$7 activation voltage had no effect on AP firing frequency (*Figure 5—figure supplement 1C*). Furthermore, shifting the supplemented K$_v$7 conductance density or its voltage dependence did not affect AP waveform properties (*Figure 5—figure supplement 1D*). Hence upon model exploration of relevant biophysical parameters, we could fully recapitulate the AP firing phenotypes observed in 5xFAD PV cells via biophysical shifts in K$_v$3 alone.

## Introduction of modified K$_v$3 conductance reproduces near-threshold hypoexcitability in PV interneurons

While powerful, model predictions are based on simplified biophysical information. To increase confidence that altered K$_v$3 channel properties can explain reduced near-threshold excitability in intact PV neurons, we employed an Arduino-based dynamic clamp system (*Desai et al., 2008*). Dynamic clamp allows real-time injection of current constrained by predefined voltage-gated conductances, such as gK$_v$3, during current clamp recordings (*Figure 6A*). Furthermore, distinct properties (e.g., activation voltage) of these conductances can be adjusted online during recordings. Dynamic clamp recordings

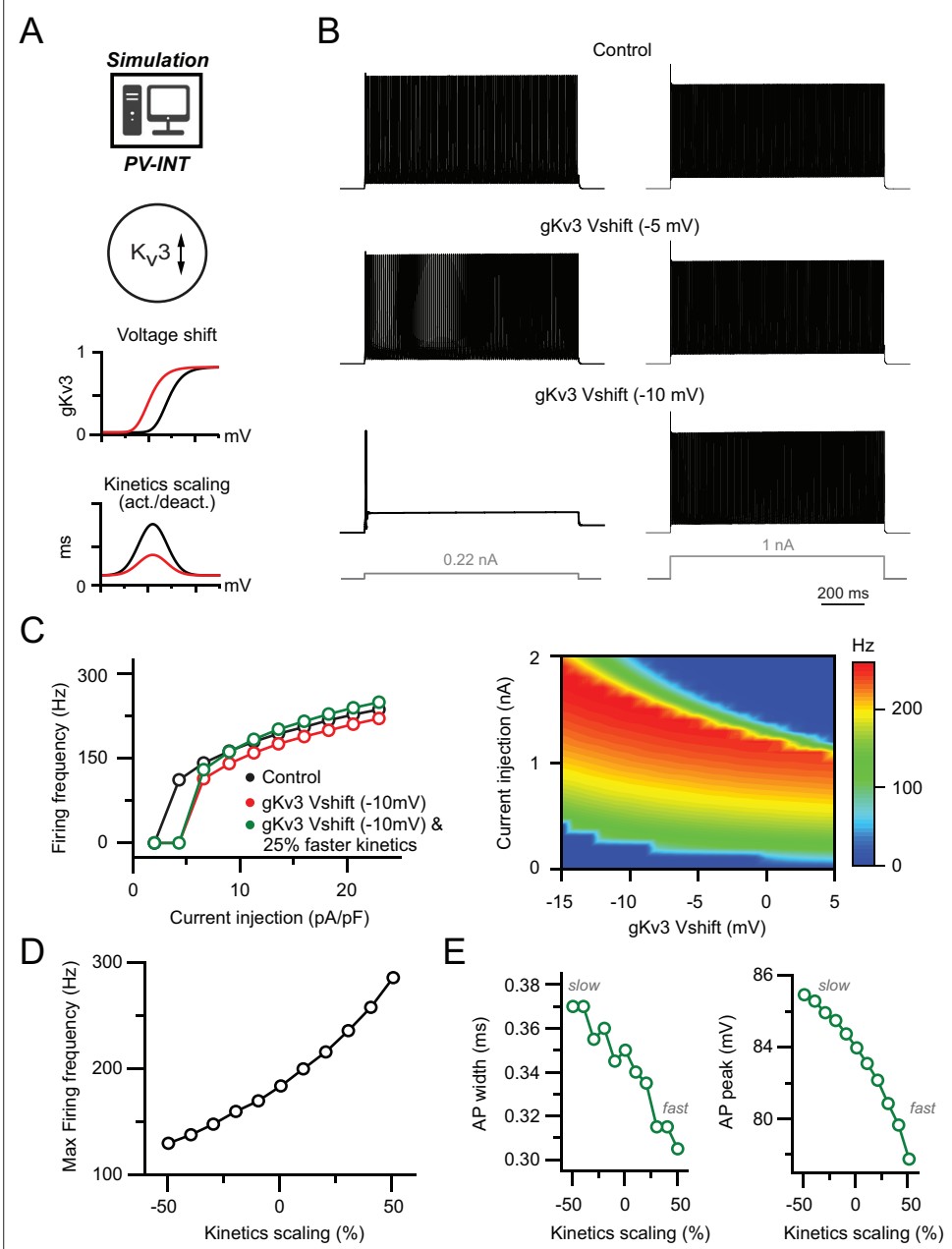

**Figure 5.** Effect of biophysical K$_v$3 dysregulation on action potential (AP) firing in a parvalbumin (PV) model. (**A**) PV cell single-compartment model with modified K$_v$3 channel properties. K$_v$3 activation voltage and kinetics were independently or simultaneously modified in the following simulations. When applied, activation and deactivation kinetics were scaled together (±50% of control). (**B**) AP firing elicited by square pulse current injections at control and hyperpolarized K$_v$3 activation voltages. Two example current injection magnitudes are displayed. (**C**) Summary of firing frequency changes in different simulated K$_v$3 conditions. Near-threshold AP firing is reduced with hyperpolarized K$_v$3 activation independent of shifting K$_v$3 activation kinetics. (**D**) Effect of modifying K$_v$3 channel activation kinetics (±50% of control) alone on maximal firing frequency in PV neuron compartmental model. (**E**) Effect on K$_v$3 channel activation kinetics changes on simulated AP width and amplitude.

The online version of this article includes the following figure supplement(s) for figure 5:

**Figure supplement 1.** Effects of supplementing different K$_v$7 conductances on action potential (AP) firing in a parvalbumin (PV) model.

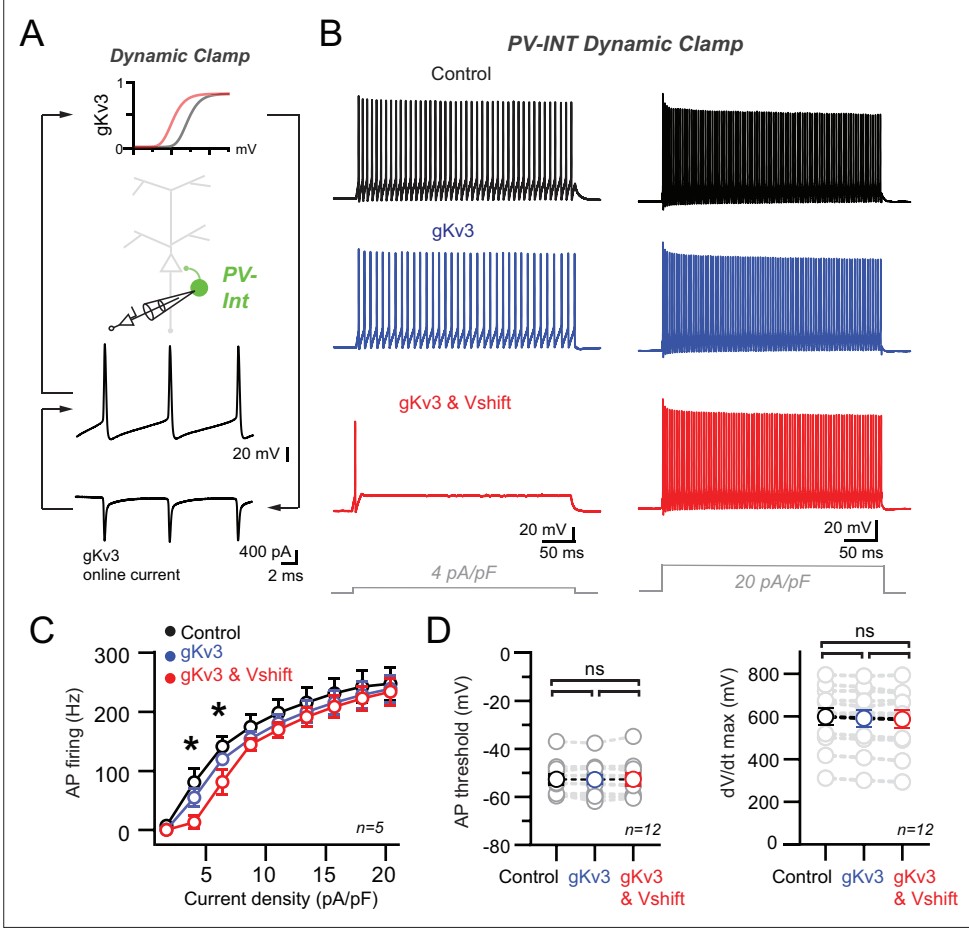

**Figure 6.** Recapitulation of the 5xFAD phenotype in parvalbumin (PV) cells using dynamic clamp. (**A**) Targeted dynamic clamp recordings from an AAV.E2.GFP⁺ neuron. Online $K_v3$ response (20 nS online $gK_v3$) shown during action potential (AP) firing in a PV interneuron. (**B**) AP firing responses to two different square pulse current injection levels in three distinct $K_v3$ dynamic clamp conditions in the same cell. (**C**) Summary data plot across a range of current injections from dynamic clamp conditions. Statistical significance was tested between the $gK_v3$ (blue) and $gK_v3$ and Vshift (red) conditions by repeated-measures (RM) two-way ANOVA ($p<0.05$) with Sidak's multiple-comparison test. (**D**) Summary plots for AP threshold and dV/dt maximum in each of the dynamic clamp conditions tested within each cell. No differences were observed in any condition using RM one-way ANOVA ($p<0.05$) with Tukey's multiple-comparison test.

The online version of this article includes the following figure supplement(s) for figure 6:

**Figure supplement 1.** Effects of wild-type (WT) and Alzheimer's disease (AD) $gK_v3$ dynamic clamp with endogenous $K_v3$ channels blocked.

were performed in targeted recordings from AAV.E2.GFP⁺ neurons in WT mice using modeled $gK_v3$ parameters described earlier. We found that dynamic clamp introduction of WT $gK_v3$ (absolute half-activation voltage, –5.0 mV) could restore fast firing after $K_v3$ blockade (*Figure 6—figure supplement 1A and B*).

To model the effect of AP firing in WT and AD-like conditions, we examined distinct $gK_v3$ conditions (*Figure 6B*; *Control* [no dynamic clamp conductance added, 0.0 nS $gK_v3$]; +$gK_v3$ [absolute half-activation voltage, –5.0 mV]; and +$gK_v3$ *and Vshift* ['5xFAD' absolute half-activation voltage, –15.0 mV]). Modest supplementation of additional *Control* $K_v3$ conductance (+$gK_v3$; 20 nS) had no discernible effect on AP firing across a range of current densities (*Figure 6B and C*). However, introduction of an identical magnitude of the 5xFAD-modeled $K_v3$ conductance (+$gK_v3$ *and Vshift*; 20 nS) induced a specific reduction in near-threshold firing without affecting high-end frequencies (*Figure 6B and C*). This +$gK_v3$ *and Vshift* induced near-threshold effect could also be replicated in 1 mM TEA (*Figure 6—figure supplement 1C and D*) following a leftward shift in the reintroduced

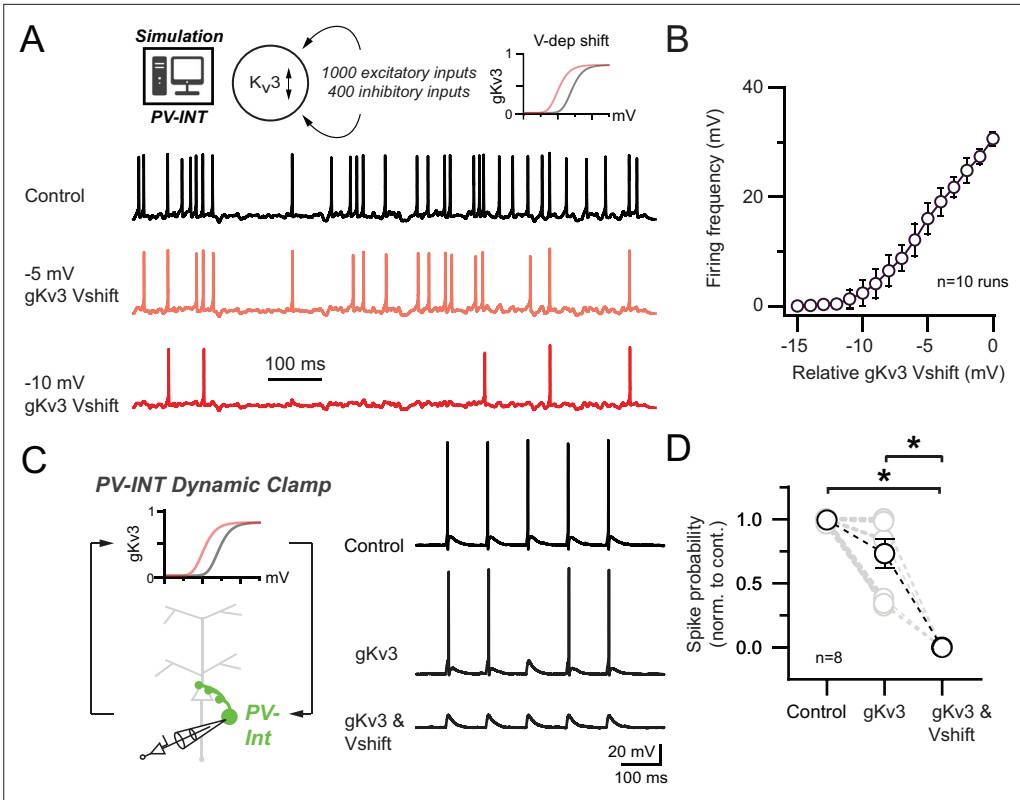

**Figure 7.** Effect of 5xFAD-related K$_v$3 channel modulation on synaptically evoked action potential (AP) firing.
(**A**) Simulated responses of parvalbumin (PV) cell compartmental model with continuous excitatory and inhibitory inputs in control and with hyperpolarized K$_v$3 activation voltages. (**B**) Summary graph of PV compartmental model firing frequency in response to continuous synaptic inputs at increasingly hyperpolarized K$_v$3 activation voltages. 0 mV represents the relative control K$_v$3 activation voltage. (**C**) 10 Hz gEPSP-evoked AP firing in dynamic clamp recordings from AAV.E2.GFP$^+$ neurons in acute slice. In control conditions, gEPSP conductance was calibrated such that the majority of stimuli evoked APs. Within recordings, the gEPSP amplitude was constant while the cell was subjected to varying gK$_v$3 dynamic clamp conditions. (**D**) Spike probability summary in response to gEPSPs in varying gK$_v$3 dynamic clamp. Significance was defined by one-way ANOVA (p<0.05) with Tukey's multiple-comparison test. For all summary graphs, data are expressed as mean (± SEM).

The online version of this article includes the following figure supplement(s) for figure 7:

**Figure supplement 1.** Effect of 5xFAD-related K$_v$3 channel modulation on synaptically evoked subthreshold events.

gK$_v$3 conductance. Compared to control, AP threshold and dV/dt maximum were unchanged in both *gK$_v$3* test conditions (*Figure 6D*). Together with our NEURON simulation data, these dynamic clamp recordings indicate that introduction of a biophysically modified K$_v$3 conductance can reproduce the hypoexcitable firing phenotype observed in PV interneurons in prodromal 5xFAD mice.

In all datasets, individual values are expressed as mean (± SEM).

## K$_v$3 modulation reduces synaptically evoked AP firing in PV interneurons

In vivo, cortical PV neurons often fire at the lower end of their dynamic range (*Yao et al., 2020*; *Yu et al., 2019*). To examine how K$_v$3 channel modulation affects PV interneuron firing in a realistic network condition, we imposed several hundred sparsely active (see 'Materials and methods') excitatory and inhibitory synapses onto our PV NEURON simulation (*Figure 7A*). In control conditions, the PV cell fired regularly (30.64 ± 0.39 Hz). Hyperpolarization of the control K$_v$3 membrane potential dependence was inversely correlated with spike frequency (*Figure 7B*).

Using dynamic clamp in WT mice, we next sought to understand whether K$_v$3 channel regulation could also diminish synaptically evoked AP firing in intact PV (AAV.E2.GFP$^+$) interneurons.

In vivo, single excitatory synaptic inputs can reliably drive AP firing in PV neurons (*Jouhanneau et al., 2018*). Thus, we injected PV neurons with an excitatory conductance (gEPSP) (*Sharp et al., 1993*, *Jaeger, 2011*, *Xu-Friedman and Regehr, 2005*) to reliably evoke AP firing at 10 Hz (gEPSP, 4.7 ± 1.0 nS; *Figure 7C*). Dynamic clamp addition of WT K$_v$3 conductance (+*gK$_v$3*; 20 nS) had a nonsignificant effect on gEPSP-evoked AP firing (*Figure 7C and D*). Interestingly, injection of the 5xFAD-modeled K$_v$3 conductance (+*gK$_v$3 and Vshift*; 20 nS) strongly reduced gEPSP-evoked firing (*Figure 7C and D*).

While often referred to as high-voltage activating channels, K$_v$3 channels open in the subthreshold range in cerebellar GABAergic interneurons (*Rowan and Christie, 2017*) and regulate the magnitude of EPSPs in hippocampal PV cells (*Hu et al., 2010*). In PV NEURON simulations, hyperpolarizing the K$_v$3 activation voltage could reduce the amplitude of EPSPs (*Figure 7—figure supplement 1A*), thus necessitating an increase in excitatory synaptic conductance to evoke an AP (*Figure 7—figure supplement 1B*). This modulation was also observed in further dynamic clamp PV recordings with subthreshold gEPSPs (3.6 ± 0.8 nS; *Figure 7—figure supplement 1C*). Together, these data argue that enhanced subthreshold activation of K$_v$3 contributes to near-threshold PV hypoexcitability during early-stage AD.

## Modulation of PV K$_v$3 channels elicits network hyperexcitability in a reduced layer 5 circuit model

Precisely timed synaptic inhibition of neuronal circuits provided by PV interneurons is indispensable for network operations (*Cardin et al., 2009*; *Da et al., 2021*; *Fuchs et al., 2007*; *Sohal et al., 2009*). In order to understand the network consequences of the observed PV phenotype in young 5xFAD mice, we developed a local PV-PC network model (*Figure 8A*). Connection strengths and probabilities for the network consisting of 200 PCs and 20 PV cells were based on previous reports (*Bock et al., 2011*; *Galarreta and Hestrin, 2002*; *Hofer et al., 2011*; *Markram et al., 2015*; *Perin et al., 2011*). The model reproduced key features of local PV circuit models including gap-junction-related firing synchrony (*Wang and Buzsáki, 1996*) and recurrent connection-related synchrony (*Bartos et al., 2007*).

We found that gradual shifting of the voltage dependence of gK$_v$3 conductance in PV cells markedly increased the firing rate of the simulated PCs (*Figure 8B*, control: 7.07 ± 0.42 Hz, 10 mV; Vshift: 30.3 ± 0.12 Hz, n = 200, p<0.0001, paired *t*-test). This network hyperexcitability can be attributed to the altered excitation–inhibition ratio due to the effects of gK$_v$3 biophysical changes of PV interneuron firing. Specifically, in the control network, PV firing (62.9 ± 6.58 Hz mean firing, n = 20) was constrained by their recurrent connections, gap junctions, and sporadic entrainment by the PC population's low firing rate. However, when the excitability of PV cells was dampened by altered gK$_v$3 voltage dependence (*Figure 8C*; n = 20 runs), PCs were released from the high inhibitory tone resulting in network hyperexcitability, which is a hallmark of recurrently connected pyramidal cells networks (*Morgan et al., 2007*, *Paz and Huguenard, 2015*).

Next, we investigated whether the increase in network excitability resulted in altered oscillatory behavior. We found that there was a significant increase in gamma power at 30 Hz (*Figure 8D*, 0.13 ± 0.08 and 38.7 ± 14.76 mV$^2$/Hz, n=5 each, p<0.05, paired *t*-test; for control and shifted gK$_v$3 network, respectively), which is in agreement with previous work (*Sohal et al., 2009*).

Our simulations demonstrate that alterations in the voltage dependence of a single PV conductance can have substantial effects on local network activity. However, minor deviations from the ensemble mean can arise from the stochastic nature of channel opening and closing (*Cannon et al., 2010*; *Lemay et al., 2011*) and from interactions with auxiliary channel subunits (*Oláh et al., 2020*; *Yu et al., 2005*). Therefore, we tested the stability of the network upon perturbations of gK$_v$3 gating. Our results showed an exponential relationship (R$^2$ = 0.93) between the voltage shift of gK$_v$3 in PV cells (*Figure 8E*) and network gamma power. This nonlinearity indicates that although an ~10 mV shift can alter circuit behavior, the network is protected against expected stochastic ion channel fluctuation-induced alterations in excitability. Together, our results demonstrate that a hypersynchronous (*Figure 8—figure supplement 1*) and hyperactive network activity can emerge as a consequence of altered PV interneuron K$_v$3 biophysics.

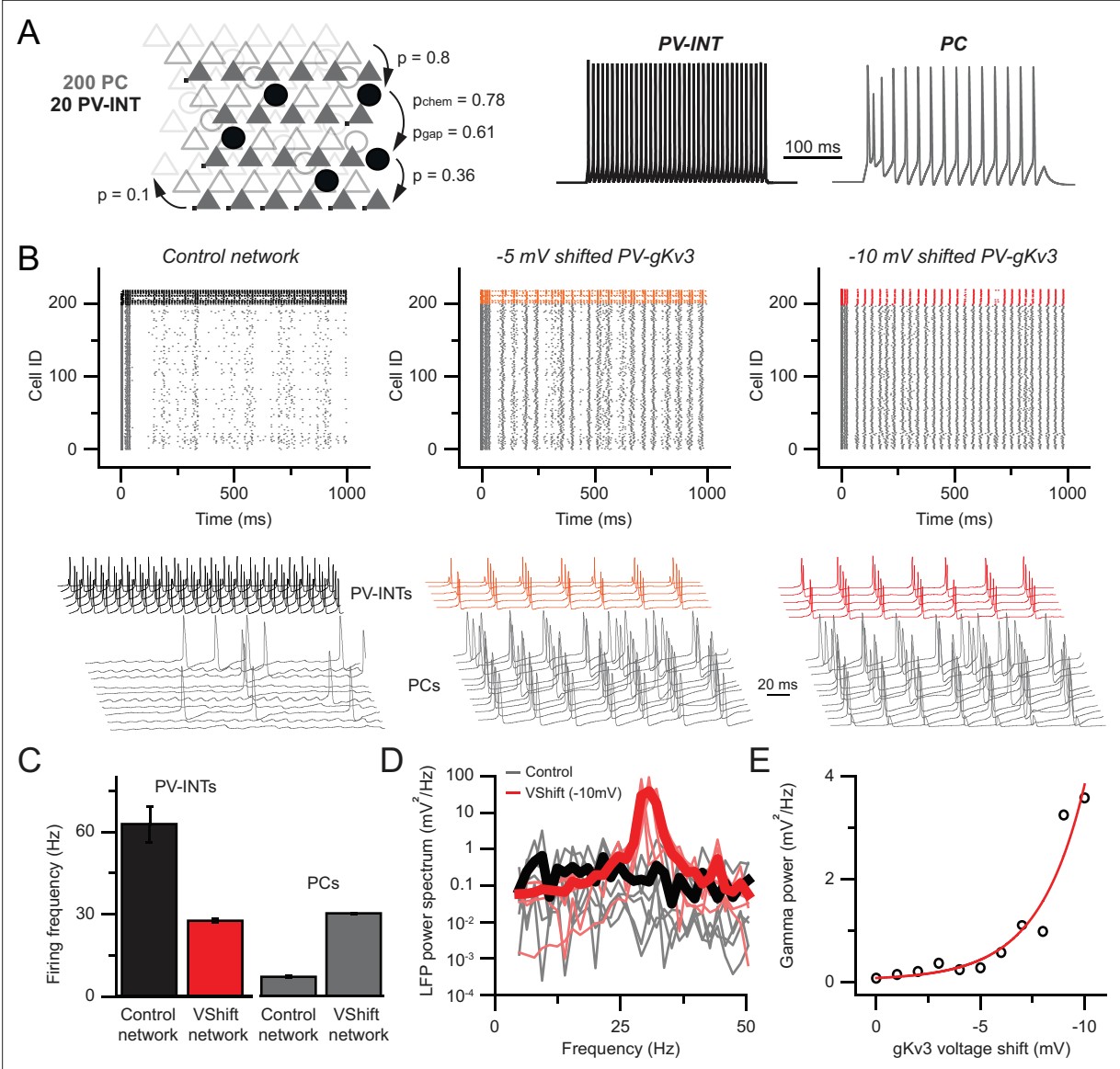

**Figure 8.** Hyperexcitability and increased gamma following parvalbumin (PV)-specific K$_v$3 modulation. (**A**) Simplified cortical network consisting of 200 pyramidal cells (PC; triangles) and 20 PV (circle) cells. Connection probabilities between and within cell groups are set based on literature. 300-ms-long spiking responses for single PC and PV cells are shown on the right. (**B**) Raster plots depicting 1-s-long network activity of the 220 cells in the network. The top 20 cells correspond to PV cells (black, orange, red), bottom 200 cells show PC activity (gray). The effect of relative –5.0 and –10.0 mV shifts in gK$_v$3 of PV cells is compared to control. Representative traces are shown from 5 PV cells and 10 PC. (**C**) Mean firing frequency of PV cells and PCs upon –10 mV relative voltage shift of gK$_v$3 in PV cells. Data are expressed as mean (± SEM). (**D**) Calculated local field potential (LFP) between 5 and 50 Hz, produced by 220 cells in the network. The activity level of individual cells was randomized and network simulations were repeated five times in control conditions and with a –10 mV relative shift in gK$_v$3 of PV cells. Individual LFP traces are shown in light gray and light red. Mean LFP traces are shown in bold black and red. (**E**) Gamma power in relation to the voltage shift of gK$_v$3 in PV cells. Gamma power was calculated by averaging LFP signals between 30 and 50 Hz. Continuous red line depicts the exponential relationship between the two variables.

The online version of this article includes the following figure supplement(s) for figure 8:

**Figure supplement 1.** Circuit synchronization following parvalbumin (PV)-specific K$_v$3 modulation.

## Discussion

In this study, we report a novel mechanism contributing to cortical circuit dysfunction in an early-stage AD mouse model. Our findings indicate that modulation of K$^+$ channel biophysics contributes to cortical PV interneuron dysfunction in early AD. In a simplified circuit model, this K$^+$ channel mechanism caused cortical network hyperexcitability and modified signaling specifically in the gamma frequency

domain. Our results represent a novel cellular mechanism with a causal link to overall circuit hyperexcitability, thus presenting a potential therapeutic avenue to combat AD progression in its early stages.

## PV interneuron pathophysiology in AD models

PV-positive GABAergic interneurons constitute a substantial proportion (~40%) of the total cortical interneuron population (*Tremblay et al., 2016*). These interneurons form powerful inhibitory synapses with local pyramidal neurons, thereby regulating a variety of cognitive functions (*Yao et al., 2020*). In several different AD mouse models, investigators have observed abnormal PV intrinsic excitability; however, mechanistic understanding of this phenomenon is incomplete. Here, we report reduced cortical PV firing in the 5xFAD model. In complementary AD mouse models, human APP and PS1 proteins (e.g., APP/PS1, hAPPJ20) are also expressed at high levels and include mutations, resulting in increased amyloid production. Within these models, PV interneurons display physiological phenotypes, including altered AP firing (*Hijazi et al., 2020*; *Verret et al., 2012*). Notably, PV neurons were found to be more susceptible to shifts in their excitability with respect to neighboring pyramidal neurons in these studies. PV-specific vulnerability could manifest as a result of their high metabolic demand (*Ruden et al., 2021*) or through abnormal regulation of ion channel subunits necessary for maintaining their fast-spiking nature (*Martinez-Losa et al., 2018*).

Related changes in PV neuron excitability are evident among the hAPP mouse models. In layer 5 PV cells, we observed reductions in near-threshold AP firing and AP width, but AP amplitude and passive properties were largely unaffected. In hippocampal CA1 from 5xFAD mice, AP firing during synaptic recruitment was also strongly reduced (*Caccavano et al., 2020*). In layer 2/3 PV neurons of hAPPJ20 mice, overall AP firing rates were unchanged but a significant reduction in AP amplitude was observed (*Verret et al., 2012*); however, in hAPPJ20 hippocampal CA1, spike frequency was strongly reduced (*Mondragón-Rodríguez et al., 2018*). A CA1 study from APP/PS1 mice observed reduction in AP width but increased AP frequency (*Hijazi et al., 2020*). In next-generation hAPP KI mice, which express the hAPP at far lower levels with respect to the aforementioned APP models, PV firing frequency was also reduced in entorhinal cortex before plaque deposition (*Petrache et al., 2019*). Variations among these studies could depend on the disease severity at which observations were made, regional differences, or genetic differences between models. Nonetheless, the related phenomena evident across these studies suggests that a unifying set of molecular mechanisms may spark circuit-level dysfunction in early AD.

## Mechanisms of altered PV excitability in AD

In a hallmark set of studies, differential expression of voltage-gated $Na^+$ channels in PV neurons was linked with network hyperexcitability in hAPP-expressing AD mice (*Martinez-Losa et al., 2018*; *Verret et al., 2012*). It is unclear whether other channel types are regulated and contribute to PV neuron dysfunction in AD. In this study, we observed physiological changes in 7–8-week-old 5xFAD mice; however, few proteomic changes are predicted until ~4 months of age in this model (*Bundy et al., 2019*). In keeping with this finding, we did not observe differences in $Na_v1$ or $K_v3$ mRNA levels in 7–8-week-old mice. However, steady-state mRNA and protein levels are not always well correlated (*de Sousa Abreu et al., 2009*; *Vogel and Marcotte, 2012*). Therefore, we compared a significant subset of the relevant cortical voltage-gated channel proteome from 5xFAD and WT mice using mass spectrometry across several ages.

In general, the number of channels showing genotype-associated changes increased with age in 5xFAD mice (*Bundy et al., 2019*). Similar to $K_v3$ mRNA, $K_v3$ protein levels ($K_v3.1$–$3.3$) were unchanged in ~7-week-old mice. Interestingly, $K_v3.3$ protein expression was reduced in more aged 5xFAD mice, displaying progressive depletion with age. Along with other $K_v3$ subunits (*Weiser et al., 1994*; *Weiser et al., 1995*), $K_v3.3$ expression is relatively high in PV neurons (*Chang et al., 2007*) and alternative splicing of $K_v3.3$ is associated with temporal lope epilepsy (*Heinzen et al., 2007*). Thus, network hyperexcitability in intermediate–late AD could be associated with altered $K_v3.3$ expression.

Unfortunately, $K_v3.4$ protein was not isolated in our mass spec analysis. As $K_v3.4$ upregulation has been shown in humans and animal models (*Angulo et al., 2004*; *Boscia et al., 2017*) or following Aβ treatment (*Pannaccione et al., 2007*), future studies should focus on evaluating regional $K_v3.4$ mRNA and protein expression in different AD models and disease stages, including well before extensive amyloid plaque deposition. Although $K_v3$ channels are highly expressed in PV cells, our proteomic

analysis was from bulk homogenates. Thus, cell-type-specific proteomic approaches in 5xFAD and other AD models should be a major focus for future work.

Rather than changes in expression levels, our results indicate that biophysical modulation of K$_v$3 channels was responsible for reduced AP firing and AP width in young 5xFAD mice. Interestingly, reduced AP width was observed in PV cells before other intrinsic alterations in APP/PS1 mice (*Hijazi et al., 2020*), suggesting that K$_v$3 modulation could precede that of other channels or homeostatic responses. Several APP-related cellular processes could explain the biophysical modulation of K$_v$3 observed here. The intermediate APP transmembrane protein product C99, produced following β-secretase (BACE1)-directed cleavage, can regulate K$_v$ channel activity (*Manville and Abbott, 2021*). In addition, increased levels of extracellular Aβ may regulate K$_v$ channel conductance either through direct interaction or via other indirect mechanisms (*Farley et al., 2021*). One or more of these APP-related interactions could contribute to the K$_v$3 channel dysregulation observed in 5xFAD mice here.

Biophysical modulation of K$_v$3 could also arise through several other well-described mechanisms without direct hAPP interactions. Changes in K$_v$3 phosphorylation via PKC, PKA, nitric oxide phosphatase (*Atzori et al., 2000*; *Beck et al., 1998*; *Desai et al., 2008*; *Kaczmarek and Zhang, 2017*; *Macica et al., 2003*; *Moreno et al., 2001*), or casein kinases (*Macica and Kaczmarek, 2001*), as well as via K$_v$3 glycosylation (*Murashov et al., 2017*), can impart changes in K$_v$3 conductance, voltage dependence, or kinetics. Future work to characterize the phosphorylation and glycosylation state of K$_v$3 in AD models will be necessary. Differential surface expression of K$_v$3 subunits or splice variants could also explain the K$_v$3 phenotype described here. For example, K$_v$3.4 subunits can increase K$_v$3 activation kinetics while also hyperpolarizing their activation voltage in cerebellar interneurons (*Baranauskas et al., 2003*; *Rowan et al., 2016*). However, of three K$_v$3.4 splice variants (K$_v$3.4a-c) only one (K$_v$3.4a) could impart these features in vitro (*Baranauskas et al., 2003*). Intriguingly, increased BACE1 activity in AD (*Rossner et al., 2006*) may promote surface expression of K$_v$3.4 subunits. BACE1 may also physically associate with K$_v$3 channel proteins in a beta-subunit-like fashion to modify their gating properties (*Hessler et al., 2015*). Additionally, changes in ancillary protein (e.g., K$_v$ beta subunit *Kcne*) expression or activity represent another avenue for modulation of K$_v$3 biophysics. For example, co-expression of K$_v$3 channels with *Kcne3* hyperpolarized their activation voltage (*Abbott et al., 2001*). While not well characterized in PV interneurons to date, *Kcne* subunits may be differentially regulated in AD (*Pannaccione et al., 2007*; *Sachse et al., 2013*). Cortical single-cell RNAseq datasets from the Allen Institute (*Gouwens et al., 2019*; *Gouwens et al., 2020*) show no expression of *Kcne*1-3 in cortical PV interneurons, and a variable level of *Kcne4* expression (our analysis). Intriguingly, the APP cleavage product C99 displays significant sequence homology with *Kcne* (*Manville and Abbott, 2021*), suggesting that K$_v$3 channels could be biophysically regulated via C99 in a similar manner as with *Kcne*. Implementing the PV-type-specific viral approach utilized in this study in various AD models will allow for a deeper evaluation of the possible mechanisms responsible for K$_v$3 modulation in future work. Additional longitudinal studies at multiple stages of the disease will be necessary to parse out the emergence of cell-type-specific biophysical mechanisms during the disease.

## Relationship of PV interneuron dysfunction and circuit-level disruptions

Circuit hyperexcitability is a prodromal indicator in familial and late-onset AD (*Dickerson et al., 2005*; *Hämäläinen et al., 2007*; *Miller et al., 2008*; *Sperling et al., 2010*; *Busche and Konnerth, 2015*; *Lamoureux et al., 2021*; *Minkeviciene et al., 2009*; *Nuriel et al., 2017*; *Quiroz et al., 2010*; *Sepulveda-Falla et al., 2012*). Altered PV interneuron firing occurs at early stages of the disease (*Hijazi et al., 2020*; *Petrache et al., 2019*), likely contributing to epileptiform activity and overall circuit hypersynchrony in cortex. Using a layer 5 cortical circuit model, we found that PV-specific K$_v$3 channel dysfunction resulted in overall hyperexcitability (*Busche et al., 2008*; *Palop and Mucke, 2016*).

Several PV cell-specific cellular and connectivity features, such as short input integration time window (*Hu et al., 2010*, *Geiger et al., 1997*), frequent recurrent connections, and extensive gap junction coupling (*Galarreta and Hestrin, 2002*), help regulate cortical circuit operations. PV cells are particularly important for maintaining signaling in the gamma frequency domain (*Bartos et al., 2007*). In our 5xFAD simulation, which produced near-threshold reduction in PV firing, we observed a sharp increase in gamma power that scaled with the severity of K$_v$3 modulation. Similarly, reduced PV excitability can amplify gamma power in different cortical areas (*Picard et al., 2019*) likely through

disruption of feedback inhibitory circuits (*Sohal et al., 2009*). Notably, increased gamma power was observed in AD patients during resting states (*Wang et al., 2017*). In the context of these studies, it is tempting to hypothesize that near-threshold changes in PV firing may disrupt inhibitory feedback circuits in cortex in times of sparse coding. Conversely, reduction of PV excitability can also result in reduced gamma power in different contexts (*Carlén et al., 2011*). Thus bidirectional, PV-specific modulation of the gamma range is likely to be circuit and context-dependent (*Sohal et al., 2009*). The tendency for local gamma power to increase or decrease in different circuits in AD should provide insight into PV-specific cellular pathology.

Further disentanglement of the mechanisms of interneuron dysfunction in distinct AD models is necessary. Specifically, the relationship of hAPP, amyloid (*Johnson et al., 2020*; *Rodgers et al., 2012*), and its intermediate products to PV-related dysfunction and abnormal circuit function. The versatility and efficiency provided by the cell-type-specific enhancer approach used here can be implemented in future studies on novel AD mouse models, or by transgene expression through viral delivery (*Kim et al., 2013*), as well as in iPSC-derived human neurons.

## Potential therapeutic strategies for amelioration of K$_v$3-related PV hypofunction in early AD

Our findings suggest an opportunity for implementation of novel targeted therapies to improve cortical circuit hyperexcitability in AD. Our biophysical, dynamic clamp, and modeling experiments here indicate that a specific K$_v$3 biophysical parameter, altered in 5xFAD mice (hyperpolarized activation voltage), can strongly alter PV firing and overall circuit activity. Our data suggest that strategies to increase expression of WT K$_v$3 are unlikely to rescue the AD firing phenotype, as supplementation of WT gK$_v$3 did not affect near-threshold PV excitability. However, drugs that depolarize the activation voltage of endogenous K$_v$3 channels, or PV-specific genetic therapies (*Vormstein-Schneider et al., 2020*) to modify K$_v$3 activation voltage dependence (*Baranauskas et al., 2003*; *Rowan et al., 2016*), present promising avenues for therapeutic intervention. Firing in our PV model was not highly sensitive to changes in other K$_v$3 properties, such as inactivation kinetics. Thus, some off-target K$_v$3 effects of pilot therapeutics may be acceptable. To better understand the translational scope of our findings, future work should focus on understanding whether biophysical K$_v$ modifications are shared across other AD models at early stages of the disease.

# Materials and methods

**Key resources table**

| Reagent type (species) or resource | Designation | Source or reference | Identifiers | Additional information |
|---|---|---|---|---|
| Strain, strain background (*Mus musculus*) | Mouse: C57B6/J | Jackson Labs | Strain# 000664 | Wild-type mouse model |
| Strain, strain background (. *musculus*) | Mouse: 5xFAD C57B6/J | Jackson Labs | Strain # 032882 | 'AD' mouse model |
| Recombinant DNA reagent | AAV.E2.GFP | Original source: Jordane Dimidschstein (MIT) | Addgene 135631 | AAV construct to transfect and express GFP in PV cells |
| Chemical compound, drug | Tetraethylammonium (TEA) | Sigma-Aldrich | Cat# 86614 | Drug used to block K$_v$3 channels |
| Chemical compound, drug | Iberiotoxin (IBTX) | Alamone Labs | Cat# STI-400 | Drug used to block BK channels |
| Software, algorithm | NEURON simulation environment | https://neuron.yale.edu/neuron/ | | Software for neuron and network simulations |
| Software, algorithm | Dynamic clamp | *Desai et al., 2008*; *Desai, 2022*, https://github.com/nsdesai/dynamic_clamp | | Software/hardware design for dynamic clamp system |

## Acute slice preparation

All animal procedures were approved by the Emory University IACUC. Acute slices from cortex were prepared from mature 5xFAD or littermate control (C57Bl/6J) mice (7–8 weeks old). Male and female 5xFAD mice and WT littermates were used for all experiments with data collected from ≥3 mice per experimental condition. Mice were first anesthetized and perfused with ice-cold cutting solution (in mM) 87 NaCl, 25 NaHO$_3$, 2.5 KCl, 1.25 NaH$_2$PO$_4$, 7 MgCl$_2$, 0.5 CaCl$_2$, 10 glucose, and 7 sucrose. Thereafter, mice were killed by decapitation and the brain immediately removed by dissection. Brain slices (300 µm) were sectioned in the coronal plane using a vibrating blade microtome (VT1200S, Leica Biosystems) in the same solution. Slices were transferred to an incubation chamber and maintained at 34°C for ~30 min and then at 23–24°C thereafter. During whole-cell recordings, slices were continuously perfused with (in mM) 128 NaCl, 26.2 NaHO$_3$, 2.5 KCl, 1 NaH$_2$PO$_4$, 1.5 CaCl$_2$, 1.5MgCl$_2$, and 11 glucose, maintained at 30.0°C ± 0.5°C. All solutions were equilibrated and maintained with carbogen gas (95% O$_2$/5% CO$_2$) throughout.

## Electrophysiology

PV neurons were targeted for somatic whole-cell recording in layer 5 region of somatosensory cortex by combining gradient-contrast video microscopy with epifluorescent illumination on custom-built or commercial (Olympus) upright microscopes. Electrophysiological recordings were obtained using Multiclamp 700B amplifiers (Molecular Devices). Signals were filtered at 6–10 kHz and sampled at 50 kHz with the Digidata 1440B digitizer (Molecular Devices). For whole-cell recordings, borosilicate patch pipettes were filled with an intracellular solution containing (in mM) 124 potassium gluconate, 2 KCl, 9 HEPES, 4 MgCl$_2$, 4 NaATP, 3 L-ascorbic acid, and 0.5 NaGTP. Pipette capacitance was neutralized in all recordings and electrode series resistance compensated using bridge balance in current clamp. Liquid junction potentials were uncorrected.

Recordings had a series resistance >20 MΩ. Membrane potentials maintained near –70 mV (–70.7 ± 1.2 and –71.3 ± 0.8 mV; WT and 5xFAD, respectively) during current-clamp recordings using constant current bias. AP trains were initiated by somatic current injection (300 ms) normalized to the cellular capacitance in each recording measured immediately in voltage clamp after breakthrough (*Taylor, 2011*) (46.9 ± 2.5 and 46.3 ± 2.9 pF, n = 21 and 19), WT and 5xFAD, respectively; p=0.89; unpaired *t*-test. For quantification of individual AP parameters, the first AP in a spike train was analyzed at 9 pA/pF for all cells. K$^+$ channel activation curves were calculated as described (*Rowan et al., 2016*) using chord conductance (g) values from current peaks and fit with a Boltzmann function. Activation time constants were obtained by fitting the rising phase of the K$^+$ current with a single exponential function.

## Intracranial viral injections

Mice were injected with AAV(PHP.eB).E2.GFP in the SBFI vibrissal region of cortex. When performing viral injections, mice were head-fixed in a stereotactic platform (David Kopf Instruments) using ear bars, while under isoflurane anesthesia (1.8–2.2%). Thermoregulation was provided by a heating plate using a rectal thermocouple for biofeedback, thus maintaining core body temperature near 37°C. Bupivacaine was subcutaneously injected into the scalp to induce local anesthesia. A small incision was opened 5–10 min thereafter and a craniotomy was cut in the skull (<0.5 µm in diameter) to allow access for the glass microinjection pipette. Coordinates (in mm from Bregma) for microinjection were X = ±3.10–3.50; Y = −2.1; α = 0°; Z = 0.85–0.95. Viral solution (titer 1 × 10$^{09}$ to 1 × 10$^{12}$ vg/mL) was injected slowly (~0.02 µL min$^{-1}$) by using a Picospritzer (0.3 µL total). After ejection of virus, the micropipette was held in place (5 min) before withdrawal. The scalp was closed with surgical sutures and Vetbond (3 M) tissue adhesive, and the animal was allowed to recover under analgesia provided by injection of carprofen and buprenorphine SR. After allowing for onset of expression, animals were sacrificed acute slices were harvested.

## Retro-orbital injection

Male and female mice were given AAV retro-orbital injections as previously described in *Chan et al., 2017*. Mice were anesthetized with 1.8–2% isoflurane. AAV(PHP.eB).E2.GFP virus was titrated to 1 × 10$^{11}$ vector genomes total and injected in C57B6/J or 5xFAD mice to label putative PV interneurons throughout cortex. As a control, PV-Cre mice (Jackson Laboratory; stock no. 008069) were injected with AAV(PHP.eB).Flex.tdTom (Addgene). Titrated virus was injected into the retro-orbital sinus of the

left eye with a 31G × 5/16 TW needle on a 3/10 mL insulin syringe. Mice were kept on a heating pad for the duration of the procedure until recovery and then returned to their home cage for 2–3 weeks until sample collection.

## Fluorescent cell picking and qPCR

Manual cell picking was performed for single-cell isolation. 12 mice (2 genotypes × 6 animals/group) were used for cell picking experiments. Acute slices (300 μm) were acquired from 5xFAD mice and their WT littermates at 7–8 weeks of age. Acute slices obtained as described above. Slices containing SBFI cortex were placed into cutting solution with 0.5 mg/mL protease (P5147-100MG, Sigma-Aldrich) for 60 min with continuous carbogen gas bubbling. Immediately after, slices were returned to room temperature cutting solution for 10 min. Slices were then micro-dissected to isolate the cortical region containing GFP⁺ or tdTom expressing cells using an epifluorescent stereoscope (Olympus SZX12). Samples were then manually triturated in cutting solution with 1% fetal bovine serum (F2442 – 50 mL, Sigma-Aldrich) into a single-cell suspension. The sample was then diluted with ~300 μL of cutting solution, dropped onto a Sylgard (DOW)-coated Petri dish, and cells were allowed 10 min to settle. The remainder of the dish was then filled with pre-bubbled cutting solution. Cells were selected using epifluorescent illumination under an inverted microscope (Olympus IX71) using a pulled borosilicate glass pipette connected to a filter-tipped stopcock. ~200 picked cells were stored in RLT buffer (Cat# 79216 – 220 mL, QIAGEN) with 1% 2-mercaptoethanol (M6250 – 100 ML, Sigma-Aldrich) at –80°C until cDNA isolation. cDNA was generated from each sample using an RNAseq library prep method. A cDNA library was created with the CellAmp Whole Transcriptome Amplification Kit (#3734, Takara Bio) to allow for real-time PCR (qPCR) to be conducted. qPCR was then conducted with the following primers: GAPDH (Mm99999915_g1, TaqMan), *Pvalb* (Mm.2766, TaqMan), *Scn1a* (Mm00450580_m1, TaqMan), *Scn8a* (Mm00488119_m1, TaqMan), *Kcnc1* (Mm00657708_m1, TaqMan), *Kcnc2* (Mm01234232_m1, TaqMan), *Kcnc3* (Mm00434614_m1, TaqMan), and *Kcnc4* (Mm00521443_m1, TaqMan). Results of qPCR were analyzed using the Common Base Method with expression normalized to GAPDH. ΔCt values were averaged between triplicate samples from each mouse.

## PV cell NEURON modeling

Computer simulations were performed using the NEURON simulation environment (versions 7.5 and 7.6, downloaded from http://neuron.yale.edu). For PV interneuron models, a single 20 μm × 20 μm compartment was created and equipped by sodium, potassium, and leak conductances. The passive background of the cell was adjusted to recreate passive membrane potential responses of whole-cell recorded PV INs for given stimulus intensities. The sodium conductance was based on the built-in Hodgkin–Huxley model of NEURON with freely adjustable sets of parameters (*Oláh et al., 2021*). The PV potassium conductance was implemented based on a previous publication (*Lien and Jonas, 2003*) constrained by our outside-out patch recordings. The steady-state activation was governed by the following equation:

$$minf = \frac{-1}{\left(1 + \exp\left(\frac{v + 5 + vshift}{12}\right)\right) + 1}$$

where v is the local membrane potential, and *VShift* is the applied voltage shift in order to adjust membrane potential dependence. The steady-state inactivation was set as follows:

$$hinf = \frac{1}{\left(1 + \exp\left(\frac{v + 30 + vshift}{10}\right)\right)}$$

The activation and deactivation time constant was defined as

$$mtau = \left(0.5 + 4 * \exp\left(-0.5 * \left(\frac{v + vshift}{25}\right)^2\right)\right) * scale$$

where scale is the parameter by which kinetics were adjusted. Inactivation time constant was set to 1000 ms or 50–1000 ms where noted in figures. Synaptic inputs for examining firing responses under more naturalistic network conditions were supplemented by using NEURON's built-in AlphaSynapse class. During the simulation (1 s), 1000 individual excitatory synapses and 500 inhibitory synapses

were added with random timing, 10 nS synaptic conductance, and 0 or –90 mV reversal potential, respectively.

In a subset of experiments, a $K_v7$ (M-current) conductance (*Sekulić et al., 2015*) was incorporated into the $K_v3$ model. Model M-currents (half-activation voltage = –27 mV) were incorporated without changes to their kinetic parameters. To calculate the effect of M-current ($I_M$), square pulse current steps were injected into the single ($K_v3$-$K_v7$) compartmental model cell, with gradually increasing amplitude. $K_v7$ conductance density was set such that noticeable changes in the firing pattern occurred, without completely abolishing spiking during current injections. In subsequent experiments, AP firing and parameters were measured with upon altered $K_v7$ conductance densities or adjusted activation voltage dependence.

## Network simulations

Network simulations were carried out with the class representation of the previously detailed PV model cell, and a newly constructed pyramidal cell (PC) mode, which was a slight modification of a bursting model cell described by earlier (*Pospischil et al., 2008*). 200 PC and 20 PV cells were used and connected with accordance to previous publications. Recurrent PC connectivity was set to 10% (*Markram et al., 2015*), PV-to-PC connectivity was set to 36% (*Packer and Yuste, 2011*), PV cell recurrent connections occurred with 78% probability, and gap junction connectivity between these cells was 61% (*Galarreta and Hestrin, 2002*). Finally, PC innervated PV cells with 80% chance (*Hofer et al., 2011*). All simulated cells received constant current injections in order to elicit baseline firing at variable frequencies. The network construction was done in several consecutive steps. First, PV cells were connected to each other with chemical synapses constrained to elicit moderate network synchronization (*Wang and Buzsáki, 1996*). Next, PV cells were connected with gap junctions, where gap junction conductance was set to a value, which could synchronize the network further. PV cells inhibited PC cells with less inputs less than 1 mV in amplitude (*Packer and Yuste, 2011*), similarly to PC to PV connections (*Hofer et al., 2011*). Firing correlations and power spectrum were analyzed in Python. All modeling-related codes will be made available upon publication.

## Dynamic clamp

The dynamic clamp system was built in-house based on a previous publication (*Desai et al., 2008*), related online available materials (http://www.dynamicclamp.com/). The equations governing the implemented gKdr were identical to those used in the NEURON model construction. Synaptic conductances were built-in predefined conductances available from http://www.dynamicclamp.com/.

## Quantitative mass spectrometry of mouse brain

Quantitative mass spectrometry was performed on whole cortex homogenates from WT (n = 43) and 5xFAD (n = 43) mice (C57BL6J-Jax genetic background, age groups spanning 1.8–14.4 months of age, including 50% females), using previously published methods (*Johnson et al., 2022*). Brain tissue was homogenized using a bullet blender and sonication, in 8 M urea lysis buffer with HALT protease and phosphatase inhibitor cocktail (Thermo Fisher). Proteins were reduced, alkylated, and then digested (lysyl endopeptidase and trypsin) followed by peptide cleanup as previously published. Tandem mass tag (TMT, 16-plex kit, A44520) peptide labeling was performed according to manufacturer's instructions, with inclusion of one global internal standard (GIS) per batch. Samples were also randomized across six TMT batches, allowing for balanced representation of age, sex, and genotype. A detailed description of this work, including methods for sample preparation, mass spectrometry work flow, and data processing, is available online (https://www.synapse.org/#!Synapse:syn27023828), and a comprehensive analysis of these data will be published separately. Raw data were processed using Proteome Discover (version 2.1) and searched against UniProt mouse database (2020). Abundances normalized to the maximum total sample reporter ion counts were transformed as $\log_2$ of the within-batch ratio over mean within each protein isoform and within each batch. Missing values were controlled to less than 50% across all batches within each isoform-specific set of measures. After confirming the presence of batch effect, this was adjusted using bootstrap regression modeling genotype, age, sex, and batch but removing covariance with batch only (*Wingo et al., 2020*) and the batch-corrected data were used for downstream analyses. Within these data (8535 proteins in total), we extracted information limited to $K^+$ and $Na^+$ channel protein subunits of relevance to this study. We contrasted

the log$_2$-transformed protein abundance means between 5xFAD and WT mice within each age group (1.8, 3.1, 6, 10.2, and 14.4 months) to identify differentially abundant proteins. If peptides mapping to separate isoforms were identified, they were quantified separately. Unadjusted $t$-test p-values (two-tailed, assuming equal variance), Benjamini–Hochberg adjusted p-values (5% false discovery rate for determination of significance), and log$_2$ fold change differences across genotype were computed.

## Stats and analysis

Custom Python scripts, Axograph, GraphPad Prism (GraphPad Software), and Excel (Microsoft) were used for analysis with values in text and figures. Statistical differences were deemed significant with α values of $p < 0.05$. Unpaired and paired $t$-tests were used for unmatched and matched parametric datasets, respectively. Where appropriate, group data were compared with one- or two-way ANOVA, and significance between groups noted in figures was determined with Tukey's or Sidak's multiple post-hoc comparison tests. Normality was determined using D'Agostino and Pearson omnibus or Shapiro–Wilk tests. Specifics for each statistical test used are given in figure legends or in the results text where data were not included in a figure.

## Acknowledgements

We sincerely thank Niraj Desai and Dan Johnston (UT Austin) for their advisement on the dynamic clamp system. A subset of AAV viral vectors were packaged in the Emory Viral Vector Core Facility. cDNA extraction and qPCR were performed with support from the Emory Integrated Genomics Core Facility. This work was supported by NIH grants R56AG072473 (MJMR), R01-MH111529 (JD), UG3MH120096 (JD); The Emory Alzheimer's Disease Research Center Grant 00100569 (MJMR); Autifony Therapeutics (MJMR); and the Simons Foundation Award 566615 (JD); with partial support from NIH grants R01NS114130 (S Rangaraju), R01AG075820 (S Rangaraju), RF1AG071587 (S Rangaraju), RF1AG071587 (NS), R01AG061800 (NS), RF1AG062181 (NS), and F32AG064862 (S Rayaprolu).

## Additional information

### Funding

| Funder | Grant reference number | Author |
| --- | --- | --- |
| National Institutes of Health | R56AG072473 | Matthew JM Rowan |
| National Institutes of Health | R01MH111529 | Jordane Dimidschstein |
| National Institutes of Health | UG3MH120096 | Jordane Dimidschstein |
| Alzheimer's Disease Research Center, Emory University | 00100569 | Matthew JM Rowan |
| National Institutes of Health | R01NS114130 | Srikant Rangaraju |
| National Institutes of Health | R01AG075820 | Srikant Rangaraju |
| National Institutes of Health | RF1AG071587 | Srikant Rangaraju Nicholas T Seyfried |
| National Institutes of Health | R01AG061800 | Nicholas T Seyfried |
| National Institutes of Health | RF1AG062181 | Nicholas T Seyfried |
| National Institutes of Health | F32AG064862 | Sruti Rayaprolu |

| Funder | Grant reference number | Author |
|--------|------------------------|--------|
| Simons Foundation | 566615 | Jordane Dimidschstein |

The funders had no role in study design, data collection and interpretation, or the decision to submit the work for publication.

## Author contributions

Viktor J Olah, Conceptualization, Data curation, Formal analysis, Investigation, Methodology, Software, Supervision, Writing – original draft, Writing – review and editing; Annie M Goettemoeller, Data curation, Formal analysis, Investigation, Methodology, Writing – review and editing; Sruti Rayaprolu, Funding acquisition, Investigation, Methodology, Resources; Eric B Dammer, Data curation, Formal analysis, Investigation, Methodology; Nicholas T Seyfried, Funding acquisition, Investigation, Methodology, Writing – review and editing; Srikant Rangaraju, Formal analysis, Funding acquisition, Investigation, Methodology, Writing – review and editing; Jordane Dimidschstein, Funding acquisition, Investigation, Methodology, Resources, Writing – review and editing; Matthew JM Rowan, Conceptualization, Data curation, Formal analysis, Funding acquisition, Investigation, Methodology, Supervision, Writing – original draft, Writing – review and editing

## Author ORCIDs

Viktor J Olah ⬡ http://orcid.org/0000-0002-2069-7525
Matthew JM Rowan ⬡ http://orcid.org/0000-0003-0955-0706

## Ethics

This study was performed in strict accordance with the recommendations in the Guide for the Care and Use of Laboratory Animals of the National Institutes of Health. All of the animals were handled according to approved Emory University institutional animal care and use committee (IACUC) protocols (#201800199). Every effort was made to reduce animal useage and to minimize suffering.

## Decision letter and Author response

Decision letter https://doi.org/10.7554/eLife.75316.sa1
Author response https://doi.org/10.7554/eLife.75316.sa2

# Additional files

## Supplementary files

• Transparent reporting form

## Data availability

We share access to our original code for simulations single cell, reduced single cell in network, and layer 5 cortical network used in this manuscript for reviewers and the public here: https://github.com/Viktor-JOlah/KDR-in-FS-PV, copy archived at swh:1:rev:bd982aa92af11d4db123ec4f824095af1473bf52. This code dataset has been made publicly available here: https://doi.org/10.5061/dryad.08kprr557. For Mass Spec data, full source data has been provided as Figure 4 - Source Data 1.

The following dataset was generated:

| Author(s) | Year | Dataset title | Dataset URL | Database and Identifier |
|-----------|------|---------------|-------------|-------------------------|
| Oláh VJ, Goettemoeller A, Dimidschstein J, Rowan MJ | 2022 | Biophysical Kv channel alterations dampen excitability of cortical PV interneurons and contribute to network hyperexcitability in early Alzheimer's | https://dx.doi.org/10.5061/dryad.08kprr557 | Dryad Digital Repository, 10.5061/dryad.08kprr557 |

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
