## [Editor Report]

Using computational modeling and dynamic clamp recordings, this work supports the concept that hyperexcitability of cortical circuits in a familial mouse model of Alzheimer's disease is caused by impairments of biophysical properties of K_v_3 channels in parvalbumin-positive cortical interneurons.

---

## [Decision Letter]

**Decision letter after peer review:**

Thank you for submitting your article "Biophysical Kv channel alterations dampen excitability of cortical PV interneurons and contribute to network hyperexcitability in early Alzheimer's" for consideration by *eLife*. Your article has been reviewed by 2 peer reviewers, and the evaluation has been overseen by a Reviewing Editor and Jeannie Chin as the Senior Editor. The following individual involved in review of your submission has agreed to reveal their identity: Bernard Attali (Reviewer #2).

Essential revisions:

1) The voltage-dependence and kinetics of Kv3 channels have been shown to be altered by their interaction with several different ancillary proteins (Kcne subunits). A change in expression of one of these subunits could potentially explain the observed results. It would be helpful to establish if there is a change in levels of mRNA or protein for any of these subunits.

2) The major piece of evidence that the 5xFAD mutation does not change the expression of K^+^ channels is that no change in overall mRNA levels could be found for any of the four Kv3 channel genes (Figure 4F). It is well established, however, that levels of mRNA do not predict levels of gene products. Thus, there could well be changes in the levels of one or more of the different Kv3 proteins that could account for the observed changes. The manuscript should be rewritten to eliminate the claim that these experiments eliminate the possibility of altered expression levels.

3) In contrast to the present findings, a number of publications have reported that there is an upregulation of Kv3.4 mRNA and protein in Alzheimer's both in humans and in animal models of the disease, as well as in cellular models. These include; Angulo et al., J. Neuronchem. 91: 547, 2004; Boda et al., J. Mol. Neurosci. 46: 606, 2012; Boscia et al., Neurobiol. Aging 54: 187-198,2017. Pannaccione et al., Mol. Pharmacology 72: 665, 2007; Pannaccione et al., J. Neurochem. 94: 572, 2005. This literature should be discussed in the context of the present findings.

4) Both the abstract and the discussion state that a "novel mechanism" is reported, specifically that "posttranslational modulation of K^+^ channel biophysics contributes to cortical PV interneuron disfunction". There is, however, no direct investigation of posttranslational modulation in this manuscript. Post-translation modifications of Kv3 channels have been investigated for close to thirty years. These channels are well documented to be regulated by several different protein kinases including PKA, PKC and casein kinase 2, as well as by glycosylation, nitric oxide, cGMP and zinc ions. At least some of these produce effects on voltage-dependence and kinetics that are very similar to those proposed for the component of K^+^ current in cortical PV+ interneurons. Surprisingly, there is not a single reference to these types of post-translational modification in the manuscript. The strong possibility that these second messenger mediated pathways could contribute to the observed effects should be acknowledged or tested.

If the authors wish to explicitly test this hypothesis there are multiple ways in which posttranslational mechanisms could be tested. For, example, a demonstration that the K^+^ currents in wild type and 5xFAD mice are identical in the presence of a pharmacological blocker or activator of one of the signaling pathways listed above would provide very strong evidence that a posttranslational modification explains the observed results. Otherwise, simply describing the way in which these mechanisms could contribute to the observed results would be warranted, but the claim that a posttranslational mechanism is involved would have to be removed.

5) The component of current that is altered in the 5xFAD mice is referred to throughout the manuscript as "Kv3 current" based on the fact that it is blocked by 1 mM TEA. While may be a good preliminary estimate it is not a rigorous test for identification as a Kv3 current. For example, the likely contribution of TEA-sensitive BK channels is dismissed because "fast spiking-interneurons do not utilize BK to shape APs." This does not, of course, dismiss the possibility that BK channels, or other K^+^ channels partially sensitive to TEA, contribute to changes in currents recorded in voltage clamp. This should be discussed or otherwise addressed.

6) At several places in the manuscript, the authors make statements about the properties of Kv3 channels in interneurons. These include "Kv3 channels open in the subthreshold range in GABAergic interneurons" (p15) and "Kv3.4 channels activate more rapidly… in interneurons." (p21). The impression is given to the reader that these statements refer to the same cortical interneurons studied in this manuscript. Most of these references, however, refer to cells in other parts of the nervous system that may have different properties from cortical PV+ interneurons. This should be clarified when citing these references.

7) Figure 4A-E shows the main parameters characterizing the 1mM TEA-sensitive K^+^ currents, which present much analogies with Kv3.4 K^+^ currents. However, some important parameters are lacking and need to be compared between WT and 5xFAD mice. Steady-state inactivation data should be provided and compared between WT and 5xFAD mice. The voltage-dependence of activation and inactivation are equally important parameters, which will define the relevant window current acting during PV interneuron firing, notably at near-threshold and are thus needed to feed the PV interneuron model more accurately. We disagree with the authors when they claim, line 245:" moderate inactivation was observed in both control and 5xFAD mice" while giving peak to steady state amplitude ratio. Inactivation of Kv3.4-like currents is very significant and important in defining the firing discharge of PV interneurons. Looking at the representative traces, one has the impression that the inactivation is more pronounced in 5xFAD mice. Thus, steady-state inactivation data will be very informative as well as the time constant of inactivation. Along this line, why in the method section for the PC-neuron modeling, line 616, the inactivation time constant was set to 1000 ms? This value is extremely high and does not reflect the currents recorded in Figure 4B. For example, Lien and Jonas (2003) set for Kv3 currents a time constant of inactivation onset of 50 ms in dynamic clamp experiments performed in hippocampal interneurons.

8) In Figure 4F, a comparison summary of KCNC1-4 (Kv3.1-Kv3.4) mRNA expression in PV interneurons provided by qPCR experiments shows no differences between WT and 5xFAD mice. This information is interesting but yet incomplete. Indeed, Kv3.4 channels express different channel isoforms (Kv3.4a-c) that are produced by alternative splicing to generate channels with different cytoplasmic COOH termini (for review, see Kaczmarek and Zhang, Physiol. Rev, 2017). It was found that co-expression of Kv3.1b with the Kv3.4a splice variant, resulted in a channel that activated at more negative potentials (Baranauskas et al., Nat. Neurosci. 2003). Furthermore, co-expression of Kv3.4 with the MIRP2 (KCNE3) β subunit was shown to produce a hyperpolarizing shift in the voltage dependence of Kv3.4 channel activation (Abbott et al., Cell, 2001). These data should be discussed in light of the qPCR experiments.

9) In Figure 6 (dynamic-clamp experiments), introduction of modified Kv3 conductance reproduces near-threshold hypo-excitability in PV interneurons. It may be more appropriate to add the modified Kv3 conductance after pharmacological block by 1 mM TEA in WT mice (see Lien and Jonas, 2003).

---

## [Author Response]

Essential revisions:1) The voltage-dependence and kinetics of Kv3 channels have been shown to be altered by their interaction with several different ancillary proteins (Kcne subunits). A change in expression of one of these subunits could potentially explain the observed results. It would be helpful to establish if there is a change in levels of mRNA or protein for any of these subunits.2) The major piece of evidence that the 5xFAD mutation does not change the expression of K^+^ channels is that no change in overall mRNA levels could be found for any of the four Kv3 channel genes (Figure 4F). It is well established, however, that levels of mRNA do not predict levels of gene products. Thus, there could well be changes in the levels of one or more of the different Kv3 proteins that could account for the observed changes. The manuscript should be rewritten to eliminate the claim that these experiments eliminate the possibility of altered expression levels.

We thank the reviewers for these helpful points. To address these issues, we have made several changes and have now added new additional data. First, we included additional references in the results and discussion, pointing out that changes in mRNA expression are not necessarily correlated with protein-level changes. Additionally, we performed mass spectrometry (MS) data of several different ion channels and regulatory subunits with any potential relevance to our observed AP firing phenotype. The MS dataset includes nearly all *Kcna*, *Kcnc*, *Kcnq*, *Kcnd*, *Kcnma1*, and *Scn* subunits, as well as regulatory *Kcnab* and *Kcnip* subunits, among several other K^+^ and Na^+^ channel proteins. Protein levels were compared between WT and 5xFAD mouse brains at 4 different timepoints from 1.8 to 14.4 months of age. Importantly, no significant changes in Kv3 channel expression were found in young 5xFAD mice, which were age-matched to qPCR and physiology experiments in the original manuscript. This data is now included and an additional supplementary figure (Figure 4—figure supplement 2) and appendix excel file, and discussed in context (p. 22-23).

We agree that changes in *Kcne* subunits are a potential mechanism for biophysical regulation of Kv3 (and other Kv) channels. However to our knowledge, there have been no specific physiological or protein expression evaluations for *Kcne* in PV interneurons to date. To evaluate if *Kcne* subunits (1-4) are expressed in PV cortical interneurons, we scraped single cell RNAseq cortical data from the Allen Institute (Gouwens, Zheng 2020 and Gouwens, Koch 2019). No *Kcne1-3* expression was noted in PV interneurons, with a variable but low level of *Kcne*4 expression identified. Unfortunately, *Kcne* protein was not detected in our additional MS evaluations. Interestingly, the APP transmembrane product C99 displays a significant sequence homology with *Kcne*, suggesting that Kv3 channels could be biophysically regulated via C99 in a similar manner as with *Kcne*. The potential for Kcne interactions as a potential mechanism for Kv3 voltage-dependence has now been added in context (p. 24) to the discussion.

3) In contrast to the present findings, a number of publications have reported that there is an upregulation of Kv3.4 mRNA and protein in Alzheimer's both in humans and in animal models of the disease, as well as in cellular models. These include; Angulo et al., J. Neuronchem. 91: 547, 2004; Boda et al., J. Mol. Neurosci. 46: 606, 2012; Boscia et al., Neurobiol. Aging 54: 187-198,2017. Pannaccione et al., Mol. Pharmacology 72: 665, 2007; Pannaccione et al., J. Neurochem. 94: 572, 2005. This literature should be discussed in the context of the present findings.

We thank the reviewers for pointing this out. This literature was now added to the discussion (p. 23), in the context of our findings on differential Kv3 expression during the progression of the disease.

4) Both the abstract and the discussion state that a "novel mechanism" is reported, specifically that "posttranslational modulation of K^+^ channel biophysics contributes to cortical PV interneuron disfunction". There is, however, no direct investigation of posttranslational modulation in this manuscript. Post-translation modifications of Kv3 channels have been investigated for close to thirty years. These channels are well documented to be regulated by several different protein kinases including PKA, PKC and casein kinase 2, as well as by glycosylation, nitric oxide, cGMP and zinc ions. At least some of these produce effects on voltage-dependence and kinetics that are very similar to those proposed for the component of K^+^ current in cortical PV+ interneurons. Surprisingly, there is not a single reference to these types of post-translational modification in the manuscript. The strong possibility that these second messenger mediated pathways could contribute to the observed effects should be acknowledged or tested.If the authors wish to explicitly test this hypothesis there are multiple ways in which posttranslational mechanisms could be tested. For, example, a demonstration that the K^+^ currents in wild type and 5xFAD mice are identical in the presence of a pharmacological blocker or activator of one of the signaling pathways listed above would provide very strong evidence that a posttranslational modification explains the observed results. Otherwise, simply describing the way in which these mechanisms could contribute to the observed results would be warranted, but the claim that a posttranslational mechanism is involved would have to be removed.

We agree and have now removed claims of a ‘posttranslational’ mechanism throughout the manuscript. In the original manuscript, we discussed the potential regulatory action of Kv3 channels via interactions with hAPP or its cleavage products. This is, as the reviewers accurately point out, only one of several other potential mechanisms acting in isolation or combination. Thus we failed to adequately describe the mechanistic scope by which our findings could arise in early AD models.

Thus the additional potential regulatory mechanisms pointed out here, including phosphorylation through several kinases, the potential for regulation through ancillary subunits, and other mechanisms, are now discussed in the context of our biophysical findings (p 23-24).

5) The component of current that is altered in the 5xFAD mice is referred to throughout the manuscript as "Kv3 current" based on the fact that it is blocked by 1 mM TEA. While may be a good preliminary estimate it is not a rigorous test for identification as a Kv3 current. For example, the likely contribution of TEA-sensitive BK channels is dismissed because "fast spiking-interneurons do not utilize BK to shape APs." This does not, of course, dismiss the possibility that BK channels, or other K^+^ channels partially sensitive to TEA, contribute to changes in currents recorded in voltage clamp. This should be discussed or otherwise addressed.

We thank the reviewers for the question. We agree with this premise and have qualified our description of the ‘Kv3 current’, where appropriate, as simply the ‘TEA-sensitive current’ when referring to currents measured in outside out patches. To address the likelihood of the TEA-sensitive current being mediated via other Kv channel type(s) (in particular, BK or Kv7 channels) we have now included additional detail and experimental results (for BK) and also an additional new supplementary figure (for Kv7; Figure 5—figure supplement 1).

With respect to BK, we now note that although the presence and functional implications of BK channels have been confirmed in fast-spiking interneurons, their localization appears confined to axonal synapses in PV interneurons (Goldberg, Rudy 2005). Pharmacological experiments aimed at exploring the effect of BK channel blockade in PV cells have confirmed that postsynaptic inhibitory potentials on downstream neurons are perturbed by BK channel blockade (Goldberg, Rudy 2005), however, no significant alteration has been found in the PV somatically recorded spiking profile (Goldberg, Rudy 2005; Erisir, Leonard 1999). To confirm this, we puffed Iberiotoxin onto outside out patches from layer 5 PV interneurons. No changes in outward conductance were identified, indicating the absence of active BK conductance in our patch recordings. This result, along with the biophysical characteristics of the currents, strongly supports the idea that the TEA (1mM) sensitive currents are highly Kv3-specific. These points have now been integrated with context in the Results section.

Another TEA-sensitive potassium current is the Kv7.2 (KCNQ) channel-mediated M-current (Otto, Wilcox 2006). The presence of this potassium channel is also associated with spike frequency adaptation (Guan, Foehring 2011). As Kv7 currents are well characterized by modeling studies, we supplemented or original Kv3-model with an additional Kv7 conductance (Sekulić et al., 2015) and confirmed that (1) M-current can have a divisive effect on the f/I curve of PV cells and (2) but that hyperpolarizing the Kv7 activation voltage had no effect on AP firing rates (in contrast to Kv3). Finally, AP waveform and threshold parameters were not affected by the presence of Kv7 at different conductance densities or voltage dependencies. We have extended our manuscript with these explanations and a new figure (Figure 5—figure supplement 1) highlighting these findings.

6) At several places in the manuscript, the authors make statements about the properties of Kv3 channels in interneurons. These include "Kv3 channels open in the subthreshold range in GABAergic interneurons" (p15) and "Kv3.4 channels activate more rapidly… in interneurons." (p21). The impression is given to the reader that these statements refer to the same cortical interneurons studied in this manuscript. Most of these references, however, refer to cells in other parts of the nervous system that may have different properties from cortical PV+ interneurons. This should be clarified when citing these references.

We thank the reviewers for pointing this out. Differences in regional location and/or cell-type identity of cells other than cortical PV referenced has now be clarified in the text where suggested.

7) Figure 4A-E shows the main parameters characterizing the 1mM TEA-sensitive K^+^ currents, which present much analogies with Kv3.4 K^+^ currents. However, some important parameters are lacking and need to be compared between WT and 5xFAD mice. Steady-state inactivation data should be provided and compared between WT and 5xFAD mice. The voltage-dependence of activation and inactivation are equally important parameters, which will define the relevant window current acting during PV interneuron firing, notably at near-threshold and are thus needed to feed the PV interneuron model more accurately.

We agree and have now updated our manuscript with inactivation voltage dependence data from PV patch recordings. Generally, Kv3 in these PV neurons displayed little inactivation at near-threshold potentials in both WT or 5xFAD mice. We performed additional modeling experiments based on our recorded observations, examining the effect of shifting both activation and inactivation voltage dependence on AP firing frequency over a range of current injections. Overall, our steady-state voltage dependence data is highly similar to earlier Kv3.3 channel findings (Weiser, Rudy 1994; Fernandez, Turner 2003) although we cannot rule out other heteromers such as Kv3.1/Kv3.4 at the soma as well (Baranauskas, Surmeier 2003). This data is now included within an additional new supplementary figure (Figure 4—figure supplement 1) and text, and these references are now noted with context in the results.

We disagree with the authors when they claim, line 245:" moderate inactivation was observed in both control and 5xFAD mice" while giving peak to steady state amplitude ratio. Inactivation of Kv3.4-like currents is very significant and important in defining the firing discharge of PV interneurons. Looking at the representative traces, one has the impression that the inactivation is more pronounced in 5xFAD mice. Thus, steady-state inactivation data will be very informative as well as the time constant of inactivation. Along this line, why in the method section for the PC-neuron modeling, line 616, the inactivation time constant was set to 1000 ms? This value is extremely high and does not reflect the currents recorded in Figure 4B. For example, Lien and Jonas (2003) set for Kv3 currents a time constant of inactivation onset of 50 ms in dynamic clamp experiments performed in hippocampal interneurons.

Our original statement of ‘moderate’ inactivation was made because of the extreme range of inactivation kinetics seen following expression of different Kv3 subunits in the literature (for example, Kv3.4 homomers ~50ms tau inactivation; Kv3.1 >1.0 sec tau inactivation). However this portion of the results has now been revised to address this concern with new data. We now report steady-state inactivation time constants in wild type and 5xFAD mice after longer step current injections. Inactivation τwas found to be quite variable both in patches from both wild type and 5xFAD, however they were not significantly different. This data is now included in new supplementary figure (Figure 4 —figure supplement 1B).

We originally hypothesized that even very fast Kv3 inactivation kinetics would not play a strong role in modulating AP firing, due to the very rapid PV-AP (~350 μs half-width at 31°C). This was also assumed because layer 5 PV cells demonstrated extremely fast firing, despite expressing Kv3 channels with clear steady-state inactivation at suprathreshold potentials. However, we recognize this was not addressed in the original manuscript (our model inactivation τ was not properly tested within the observed range of inactivation kinetics). We now examined the effect of differing inactivation kinetics on AP firing in our PV model. Unlike shifting Kv3 activation kinetics alone (Figure 5D) which had a strong effect on AP firing, changing Kv3 inactivation kinetics did not affect AP firing at either near-threshold or high-end firing rates. This was true across a wide range of inactivation time constants. The lack of effect of fast Kv3 inactivation on AP firing is potentially due to insignificant accumulation of inactivation during very short PV-APs. It is also worth noting that changing Kv3 *activation* kinetics affected AP width (Figure 5E) whereas changing Kv3 *inactivation* (τ range tested, 50ms^-1^000ms) kinetics did not. Thus, increased use-dependent inactivation of Nav channels occurs with changes in Kv3 activation kinetics but not inactivation kinetics in this cell type. Lastly, the near-threshold dampening of AP firing seen following hyperpolarization of Kv3 activation voltage dependence was found to occur irrespective of Kv3 inactivation kinetics. This data is now included in the new supplementary figure (Figure 4—figure supplement 1).

8) In Figure 4F, a comparison summary of KCNC1-4 (Kv3.1-Kv3.4) mRNA expression in PV interneurons provided by qPCR experiments shows no differences between WT and 5xFAD mice. This information is interesting but yet incomplete. Indeed, Kv3.4 channels express different channel isoforms (Kv3.4a-c) that are produced by alternative splicing to generate channels with different cytoplasmic COOH termini (for review, see Kaczmarek and Zhang, Physiol. Rev, 2017). It was found that co-expression of Kv3.1b with the Kv3.4a splice variant, resulted in a channel that activated at more negative potentials (Baranauskas et al., Nat. Neurosci. 2003). Furthermore, co-expression of Kv3.4 with the MIRP2 (KCNE3) β subunit was shown to produce a hyperpolarizing shift in the voltage dependence of Kv3.4 channel activation (Abbott et al., Cell, 2001). These data should be discussed in light of the qPCR experiments.

These findings have now been incorporated, in context, within the discussion (p. 24).

9) In Figure 6 (dynamic-clamp experiments), introduction of modified Kv3 conductance reproduces near-threshold hypo-excitability in PV interneurons. It may be more appropriate to add the modified Kv3 conductance after pharmacological block by 1 mM TEA in WT mice (see Lien and Jonas, 2003).

We initially chose to perform dynamic clamp experiments without Kv channel blockade to evaluate how introduction of a left-shifted activating (‘AD’) gKv3 or additional wild type gKv3 would affect firing, with endogenous wild-type channels also still available. Introduction of a quite modest magnitude of left-shifted ‘AD’ gKv3 conductance in these conditions could replicate the firing phenotype of AD neurons (Figure 6B). Thus despite the availability of right-shifted endogenous channels, introduction of left-shifted gKv3 will still diminish near-threshold firing to similar inputs. Overexpression of right-shifted Kv3 channels is therefore unlikely to rescue the AD firing phenotype. However drugs which modify the voltage dependence of Kv3, or modulation of Kv3 subunit stoichiometry, present promising avenues for therapeutic intervention. We failed to adequately communicate the logic of our experimental design and these potential interpretations this in the original manuscript. We have now added additional commentary/ interpretation related to this experiment in the discussion (p. 25).

Although we consider our initial experiments to be informative/appropriate, in Lien and Jonas 2003; e.g. Figure 5, the authors demonstrated altered neuronal firing following introduction of gKv3 both in conditions where endogenous Kv channels were blocked or unblocked. Thus, we agree with reviewers that this experiment was important to perform, in addition to our initial dynamic clamp experiments without Kv3 blockade. We first found that dynamic clamp re-introduction of our ‘wild type’ gKv3 PV conductance could substantially restore firing after Kv3 blockade with TEA. Furthermore, the AD firing phenotype could also be replicated in 1mM TEA (Figure S6C,D) following a left-ward shift in the re-introduced gKv3 activation voltage. These additional experiments are now included in an additional figure (Figure 6—figure supplement 1).